# Evaluating the Robustness of Analogical Reasoning in GPT Models

**Martha Lewis**                                                              *m.a.f.lewis@uva.nl*
*ILLC, University of Amsterdam, Amsterdam, The Netherlands*
*Santa Fe Institute, 1399 Hyde Park Road, Santa Fe, NM 87501, USA*

**Melanie Mitchell**                                                              *mm@santafe.edu*
*Santa Fe Institute, 1399 Hyde Park Road, Santa Fe, NM 87501, USA*

**Reviewed on OpenReview:** *https: // openreview. net/ forum? id= t5cy5v9wph*

## Abstract

Large language models (LLMs) have performed well on several reasoning benchmarks, including ones that test analogical reasoning abilities. However, there is debate on the extent to which they are performing general abstract reasoning versus employing shortcuts or other non-robust processes, such as ones that overly rely on similarity to what has been seen in their training data. Here we investigate the robustness of analogy-making abilities previously claimed for one prominent class of LLMs—GPT models—on three of four domains studied by Webb et al. (2023): letter-string analogies, digit matrices, and story analogies. For each of these domains we test humans and GPT models on robustness to variants of the original analogy problems—versions that test the same abstract reasoning abilities but that are likely dissimilar from tasks in the pre-training data. The performance of a system that uses robust abstract reasoning should not decline substantially on these variants.

On simple letter-string analogies, we find that while the performance of humans remains high for two types of variants we tested, the GPT models' performance declines sharply. This pattern is less pronounced as the complexity of these analogy problems is increased, as both humans and GPT models perform poorly on both the original and variant problems requiring more complex analogies. On digit-matrix problems, we find a similar pattern but only on one out of the two types of variants we tested. Lastly, we assess the robustness of humans and GPT models on story-based analogy problems, finding that, unlike humans, the performance of GPT models are susceptible to answer-order effects, and that GPT models also may be more sensitive than humans to paraphrasing.

This work provides evidence that, despite previously reported successes of GPT models on zero-shot analogical reasoning, these models often lack the robustness of zero-shot human analogy-making, exhibiting brittleness on most of the variations we tested. More generally, this work points to the importance of carefully evaluating AI systems not only for accuracy but also robustness when testing their cognitive capabilities.

Code, data, and results for all experiments is available at `https://github.com/marthaflinderslewis/robust-analogy`.

**Keywords:** Analogy; Reasoning; Large Language Models; Evaluation; Robustness; Counterfactual Tasks

## 1  Introduction

The degree to which pre-trained large language models (LLMs) can reason—deductively, inductively, analogically, or otherwise—remains a subject of debate in the AI community. Many studies have shown that

LLMs perform well on certain reasoning benchmarks (Huang & Chang, 2022; Wei et al., 2022a;b). However, other studies have questioned the extent to which these systems are able to reason abstractly, as opposed to relying on shortcuts Branco et al. (2021); Taghanaki et al. (2024) or other heuristics, including "approximate retrieval" from encoded training data (Kambhampati, 2023). Several groups have shown that LLMs' performance on reasoning tasks degrades, in some cases quite dramatically, on versions of the tasks that vary from standard benchmarks or that are likely to be rare in the LLMs' training data (Dziri et al., 2023; McCoy et al., 2024a;b; Mirzadeh et al., 2024; Razeghi et al., 2022; Srivastava et al., 2024; Wu et al., 2023). The lack of robustness to variations on tasks means that the performance of LLMs on real-world tasks might not be well-predicted from their performance on standard benchmarks.

In this paper, we evaluate the robustness of analogical reasoning in three of OpenAI's GPT models. In particular, we evaluate the robustness of results reported by Webb, Holyoak, and Lu (2023) (hereafter referred to as WHL), who carried out experiments to assess GPT-3's zero-shot ability to solve analogy problems in four domains: four-term verbal analogies, digit matrices, letter-string analogies, and story analogies. They found that on several types of problems GPT-3 matched or exceeded human performance, and concluded that "GPT-3 appears to display a [zero shot] emergent ability to reason by analogy."

After repeating the experiments of WHL on letter-string analogies, digit-matrix problems, and story analogies, we assess the robustness of analogical reasoning in both humans and GPT models by testing on variants of the original tasks that are unlikely to be similar to reasoning tasks seen in the models' training data. If an LLM (or human solver) is using robust abstract reasoning procedures, it should perform comparably on both the original tasks and variants; if it is using procedures that rely on shortcuts or similarity to training data, the performance should drop substantially on the variants.[1]

For letter-string problems we tested two types of variants using "fictional alphabets"(1) in which the positions of some letters are perturbed; and (2) in which letters are replaced by non-letter symbols. On simple problems we find that while human performance remains high across variants, the GPT models' performance declines sharply. This pattern is less pronounced as the complexity of these analogy problems is increased, as both humans and GPT models perform poorly on both original and variant problems requiring more complex analogies.

For digit-matrix problems we also tested two types of variants: (1) versions in which the position of the "blank" (missing answer) in the matrix was randomly chosen rather than always being the bottom-right entry; and (2) versions in which digits were replaced by non-digit symbols. For variants of type 1 we found that while human accuracy did not change from that on the original problems, the GPT models' performance again declines sharply. For variants of type 2 we found that neither the performance of humans or GPT models changed substantially from that on the original problems.

Finally, we evaluate the robustness of humans and GPT models on story-analogy problems in two ways: we examine (1) the effects of different ordering of the answer candidates and (2) the effects of paraphrased versions of stories. We find that, unlike humans, the performance of GPT models show strong answer-order effects, and that these models seem more sensitive than humans to paraphrasing effects.

Because the purpose of this paper is to evaluate the robustness of published claims of zero-shot analogical reasoning with simple prompts, we do not experiment with other prompting formats, such as multi-shot chain-of-thought prompting, or with models specifically trained for reasoning capabilities (e.g., OpenAI (2024)), but rather leave such evaluations for future work. However, the zero-shot, simple prompt case is arguably the most interesting one, as noted by WHL: "Of particular interest is the ability of these models to reason about novel problems zero-shot, without any direct training. In human cognition, this capacity is closely tied to an ability to reason by analogy."

**Contributions:** This work provides evidence that, despite previously reported successes of GPT models on analogical reasoning, these models often lack the zero-shot robustness of human analogy-making, exhibiting brittleness on most of the variations we tested. More generally, this work points to the importance of carefully evaluating AI systems not only for accuracy but also robustness when testing their cognitive capabilities.

---

[1]We did not experiment on WHL's four-term verbal analogies; the ability to solve such "proportional analogies" based on single words comes easily to GPT models likely due to properties of the word embeddings they learn Chiang et al. (2020).

**Data Availability:** Code, data, and results for all experiments are available at `https://github.com/marthaflinderslewis/robust-analogy`.

## 2 Letter-String Analogies

### 2.1 Background

Letter-string analogies were proposed by Hofstadter (1985) as an idealized domain in which processes underlying human analogy-making could be investigated. The following is a sample problem:

$$a \; b \; c \; d \rightarrow a \; b \; c \; e \;\; ; \;\; i \; j \; k \; l \rightarrow \;\; ?$$

Here, a b c d → a b c e is called the *source transformation* and i j k l is called the *target*. The solver's task is to generate a new string that transforms the target analogously to the source transformation. There is no single correct answer to such problems, but there is typically general agreement in how humans answer them. For example, for the problem above, most people answer i j k m, and answers that deviate from this tend to do so in particular ways (Mitchell, 1993).

In addition to the work of Hofstadter & Mitchell (1994) on creating computer models of analogy-making using this domain, letter-string analogies have been used to isolate the neural correlates of analogy formation (Long et al., 2015; Geake & Hansen, 2010), and to model higher-order analogy retrieval (Dekel et al., 2023).

WHL compared the ability of GPT-3 with that of humans on a dataset of letter-string analogies, finding that, in most cases, GPT-3's performance exceeded the average performance of the human participants. Here, performance is measured as fraction of correct answers on a given set of letter-string problems, where WHL used their intuitions to decide which answer displays abstract analogical reasoning and thus should be considered "correct." In this paper we will use their definition of correctness.

WHL's dataset consists of a set of problem types involving different kinds of transformations and levels of generalization. The following are the six transformation types with sample transformations:

1. Extend Sequence: a b c d → a b c d e

2. Successor: a b c d → a b c e

3. Predecessor: b c d e → a c d e

4. Remove Redundant Letter: a b b c d → a b c d

5. Fix Alphabetic Sequence: a b c w e → a b c d e

6. Sort: a d c b e → a b c d e

Each type of transformation can be paired with a "zero-generalization" target (e.g., i j k l) or with between one and three of the following types of generalizations:

1. Letter-To-Number: a b c d → a b c e ; 1 2 3 4 → ?

2. Grouping: a b c d → a b c e ; i i j j k k l l → ?

3. Longer Target: a b c d → a b c e ; i j k l m n o p → ?

4. Reversed Order: a b c d → a b c e ; l k j i → ?

5. Interleaved Distractor: a b c d → a b c e ; i x j x k x l x → ?

6. Larger Interval: a b c d → a b c e ; i k m o → ?

The following are examples of (1) a 1-generalization problem (Extend Sequence paired with Longer Target); (2) a 2-generalization problem (Remove Redundant Letter paired with Letter-To-Number and Grouping); and (3) a 3-generalization problem (Successor paired with Grouping, Reversed Order, and Interleaved Distractors).

1. a b c → a b c d  ;  i j k l m n →  ?.

2. a b b c → a b c  ;  1 1 2 2 3 3 3 3 4 4 →  ?.

3. a b c → a b d  ;  l l x k k x j j x i i x →  ?.

Finally, WHL included a number of problems involving "real-world" concepts, such as,

$$a \ b \ c \rightarrow a \ b \ d \ ; \ \text{cold cool warm} \rightarrow \ ?$$

In this paper we do not include the "real-world" concepts in our studies, focusing instead on non-linguistic inputs.

WHL generated numerous problems for each problem type and presented these to GPT-3 (text-davinci-003) as well as 57 UCLA undergraduates. The human participants exhibited a large variance in accuracy, but on average, WHL found that GPT-3 outperformed the average human performance on most problem types.

## 2.2  Variants of Letter-String Analogies

To assess the robustness of WHL's results on letter-string analogies, we created variants of the same letter-string analogy problem types, and evaluated both humans and three GPT models on these versions.

**Fictional Alphabets**  We created two types of variants using "fictional" alphabets: problems using permuted alphabets and problems using "alphabets" of symbols rather than letters. To create a permuted alphabet, we reorder $n$ letters, where $n$ can be 2, 5, 10, or 20. For each of the four values of $n$, we generated seven distinct alphabets $\alpha_n(i)$, $i \in \{1, ..., 7\}$, with $n$ randomly chosen letters reordered, yielding $4 \times 7 = 28$ distinct permuted alphabets. We also generated two symbol alphabets, $\psi_{10}(1)$ and $\psi_{10}(2)$, each consisting of 10 non-letter symbols in a given order, and seven symbol alphabets $\psi_{15}(i)$, $i \in \{1, ..., 7\}$ each consisting of 15 non-letter symbols in a given order.

**Creating Zero-Generalization Letter-String Problem Variants**  For the zero-generalization case, for each $\alpha_n(i)$, we created 10 different letter-string problems for each of WHL's six transformation types. This results in $7 \times 10 \times 6 = 420$ zero-generalization analogy problems for each value of $n$. We added to this 420 letter-string problems using the non-permuted ($n = 0$) alphabet, spread evenly over the six transformation types. Figure 1a gives an example of a Fix Alphabetic Sequence problem using an alphabet with two letters (e and m) reordered.

For each of the symbol alphabets $\psi_{10}(1)$ and $\psi_{10}(2)$ we created 10 problems each for the Successor and Predecessor problem types, for a total of 40 distinct non-letter symbol problems. Figure 1b gives an example of a Predecessor problem using a symbol alphabet.

**Creating Letter-String Problem Variants With Generalizations**  To create variants of letter-string problems with generalizations, we used the same alphabets $\alpha_n(i)$, as well as the seven longer symbol alphabets, $\psi_{15}(i)$.

Recall that WHL created problems that paired each transformation type with either one, two, or three generalization types.

To generate 1-generalization problems with permuted alphabets, for each $\alpha_n(i)$, $i \in \{1, ..., 7\}$, and for each of the six generalization types, we created 10 problems, each with a randomly chosen transformation type. This yields $7 \times 6 \times 10 = 420$ distinct 1-generalization permuted-alphabet problems. We added to this 420 distinct 1-generalization problems for the original ($n = 0$) alphabet, spread evenly over the six transformation types.

To generate 1-generalization problems with symbol alphabets, for each $\psi_{15}(i)$, $i \in \{1, ..., 7\}$, and for each of the six generalization types, we created 10 problems, each with a randomly chosen transformation type, for a total of $7 \times 6 \times 10 = 420$ distinct 1-generalization symbol-alphabet problems.

For two- (three-) generalization problems, for each number of letters permuted and each alphabet, we create 70 problems with two (three) randomly selected generalization types and a randomly selected task. We thus create $7 \times 70 = 490$ unique two-generalization problems and the same number of unique three-generalization problems. Due to the difficulty of two- and three-generalization symbol problems for both humans and GPT models, we limited our human and GPT experiments to symbol problems only with zero- and one-generalizations.

## 2.3 Methods For Experiments On Letter-String Analogies and Variants

**Human Study Methods**   For all human studies, we collected data on Prolific Academic.[2] Participants were screened to have English as a first language, to currently be living in the UK, the USA, Australia, New Zealand, Canada, or Ireland, and to have a 100% approval rate on Prolific.

In order to assess humans' abilities on the original and variants of letter-string problems, we collected data from 263 participants. Each participant was asked to solve 14 letter-string analogy problems drawn from original, permuted, and symbol alphabets, with different numbers of generalizations. The 14 problems given to each participant were sampled from different transformation types.

In addition to the 14 problems, participants were also given two attention-check questions at random points during the experiment, with a warning that if the attention checks were failed, then payment ($7 for the experiment, which was expected to take about 20–30 minutes) would be withheld. Figure 1c gives an example of an attention check. Four of the 263 participants' submissions were rejected due to failed attention checks. As in WHL's studies, as part of the initial instructions, participants were given a simple example problem to complete and then were shown the solution.

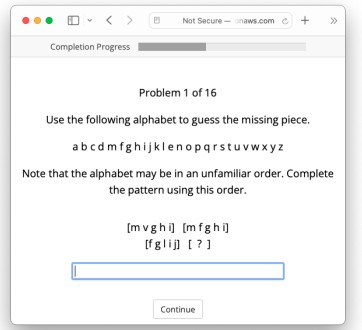 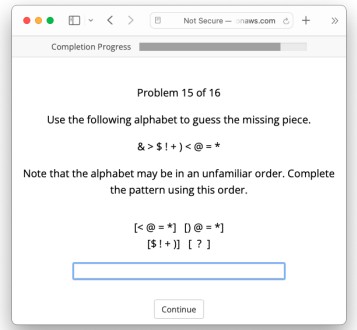 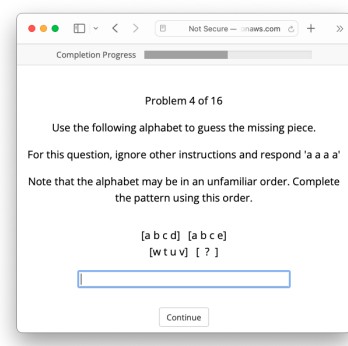

(a) Example analogy problem with permuted alphabet.

(b) Example analogy problem with symbol alphabet.

(c) Example attention check.

Figure 1: Example items in human study.

**GPT Study Methods**   We evaluated the performance of three GPT models—GPT-3 (text-davinci-003, which was the model tested by WHL), GPT-3.5 (gpt-3.5-turbo-0613), and GPT-4 (gpt-4-0613)—on the same problems given to humans. Following WHL, all GPT experiments were done with temperature set to zero. GPT-3 takes in a single prompt, whereas GPT-3.5 and GPT-4 take in a list of messages that define the role of the system, input from a "user" role, and optionally some dialogue with simulated responses from the model given under the role "assistant."

---

[2] https://www.prolific.com/academic-researchers

For our experiments on the original letter-string problems used by WHL, our system and user prompts for GPT-3.5 and GPT-4, have the following format:

> **System:** You are able to solve letter-string analogies.
> **User:** Let's try to complete the pattern:\n\n[a b c d] [a b c e]\n[i j k l] [

The user prompt is identical to the prompt WHL gave to GPT-3; the \n character signifies a line break to the model.

For our experiments on letter-string analogy variants, we tested three different prompt formats, including one similar to instructions given in our human study and one in a zero-shot chain-of-thought setup. The best performance across models was achieved with the prompt format used in Hodel & West (2023):

> **System:** You are able to solve letter-string analogies.
> **User:** Use this fictional alphabet: [a u c d e f g h i j k l m n o p q r s t b v w x y z]. \nLet's try to complete the pattern:\n[a u c d] [a u c e]\n[i j k l] [

Appendix A.1.2 gives the other prompt variants we tested, and Appendix A.2.1 gives results from the zero-shot chain-of-thought experiments.

Note that in our studies, the "fictional alphabet" part of the prompt and the alphabet listing was included even for problems using the non-permuted ($n = 0$) alphabet.

In this and all other experiments, we tested GPT-3 with a concatenation of the system and user prompts. Following WHL, in our experiments all GPT model responses were truncated at the point where a closing bracket was generated.

## 2.4 Replication of WHL's Studies

Our first set of experiments attempted a replication of WHL's experiments testing humans and GPT-3 on letter-string analogies.

**Replication: Human Study Results**  Figure 2 compares the results of our human study with that of WHL on the original letter-string problems, averaged across transformation types for different numbers of generalizations. The participants in our study achieved higher average accuracy than those of WHL on zero-generalization problems, and similar accuracy on problems with one or more generalizations. The differences on zero-generalization problems may be due to differences in experimental protocols or in the participant pools.

**Replication: GPT Study Results**  Figure 3 shows GPT-3 data from WHL compared with data from our computational experiments with GPT-3, GPT-3.5 and GPT-4, averaged across transformation types for different numbers of generalizations. In all cases our GPT-3 results are similar to those of WHL. GPT-3.5 and GPT-4 show slightly lower accuracy than GPT-3, possibly due to their fine-tuning beyond a strict prompt-completion objective.

In summary, our human and GPT-model replication results are generally consistent with those of WHL, although our human study yields higher human performance on zero-generalization problems.

## 2.5 Results on Variants of Letter-String Problems

**Counterfactual Comprehension Check On Fictional Alphabets**  For problems involving permuted alphabets, we follow Wu et al. (2023) by providing counterfactual comprehension checks (CCCs) to test if the models grasp the basics of the task proposed. We use two CCCs: firstly, given an alphabet and a sample letter (or symbol) from that alphabet, give the successor of that letter. Secondly, we use the same format but ask for the predecessor of the letter. We ensure that we do not ask for the successor of the last letter in the alphabet or the predecessor of the first. Details of our CCCs are given in Appendix A.1.1. In summary,

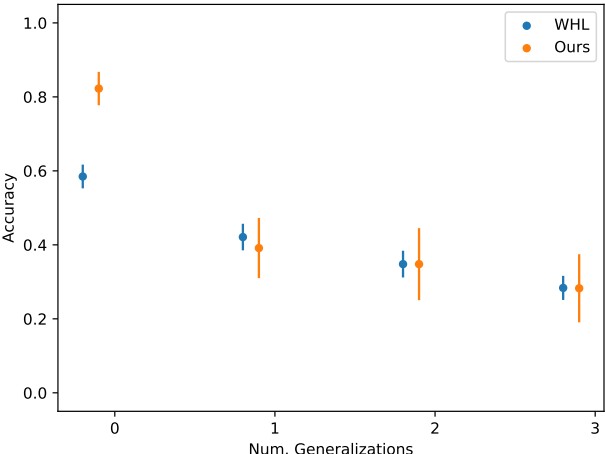

Figure 2: Human results on WHL's original letter-string task for problems with zero to three generalizations. "WHL" refers to the results of WHL's original human studies, and "Ours" refers to the results of our human studies. Data points give mean accuracy across all transformation types, and bars indicate 95% binomial confidence intervals. Numbers of samples for WHL are 342 for each number of generalizations. Numbers of samples for our data are 276 for zero generalizations, 138 for one, and 92 each for two and three generalizations. Figure is best viewed in color.

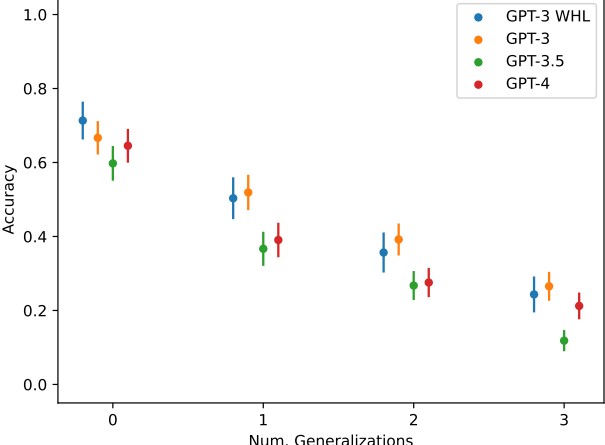

Figure 3: GPT results on WHL's original letter-string task for problems with zero to three generalizations. "GPT-3 WHL" refers to WHL's results on GPT-3. "GPT-3", "GPT-3.5", and "GPT-4" refer to our results with those models. Data points give mean accuracy across all task types, and bars indicate 95% binomial confidence intervals. Numbers of samples for GPT-3 WHL are 300 for each number of generalizations. Numbers of samples for our data are 420 for zero and one generalization, and 490 for each of two and three generalizations. Figure is best viewed in color.

we find that the CCC accuracy is generally quite high, indicating that the models generally grasp what "successor" and "predecessor" mean in each fictional alphabet.

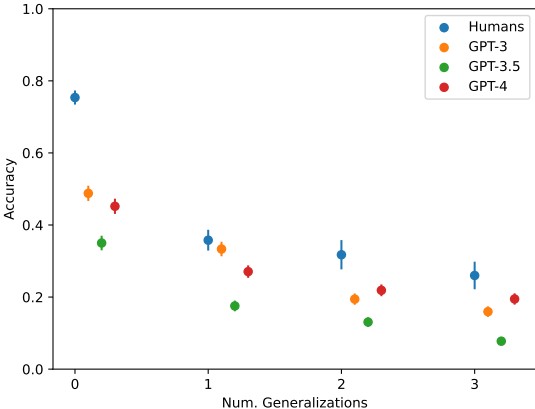

Figure 4: Accuracy for humans and GPT models in our studies, versus number of generalizations. Data points indicate mean accuracy across all alphabets and problem types and bars indicate 95% binomial confidence intervals. Number of samples for GPT models for zero generalizations is 2,140, for humans 1,876. Number of samples for GPT-3 for one generalization is 2,100, for GPT-3.5 and GPT-4 is 2560, and for humans is 1,062. Number of samples for GPT models for two and three generalizations is 2,450, and for humans is 504. Note that mean accuracies for 2 and 3 generalizations do not include symbol alphabets, and mean accuracies for GPT-3 1-generalization also do not include symbol alphabets. Figure is best viewed in color.

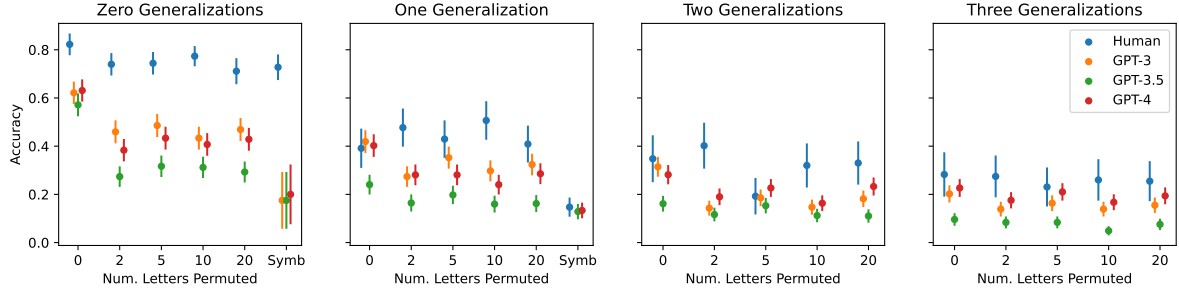

Figure 5: Accuracy of human participants and GPT models versus different alphabet types (number of permuted letters), for different numbers of generalizations. . Data points indicate mean accuracy across all transformation types for different alphabets, and bars indicate 95% binomial confidence intervals. The number of samples for each human data point is given in Table 1. The number of samples for GPT-model data point is as follows. For zero or one generalizations, and 0-20 letters permuted, each data point corresponds to 420 samples. For two or three generalizations, 0-20 letters permuted, each data point corresponds to 490 samples. For the symbol alphabets, zero-generalization data points correspond to 40 samples and one-generalization correspond to 420 samples. Figure is best viewed in color.

**Results On Human and GPT Experiments on Variants of Letter-String Analogy Problems**
Figure 4 shows the mean accuracy and 95% binomial confidence intervals for humans and GPT models across all alphabets and problem types. In all cases the average human performance on these problems is higher than that of any of the GPT models, and in the case of zero generalizations it is substantially higher. Note that GPT-3 was not available for our experiments on symbol alphabets with one generalization. Note also that due to funding limitations, we did not test humans on two- and three-generalization problems for symbol alphabets, so for both humans and GPT models, we only report results on two- and three-generalization

Table 1: Number of human samples per data point in Figure 5.

| Num. Generalizations | Alphabets | | | | | |
|:---:|:---:|:---:|:---:|:---:|:---:|:---:|
| | 0 | 2 | 5 | 10 | 20 | Symb |
| 0 | 276 | 342 | 336 | 384 | 270 | 268 |
| 1 | 138 | 153 | 156 | 150 | 159 | 300 |
| 2 | 92 | 102 | 104 | 100 | 106 | – |
| 3 | 92 | 102 | 104 | 100 | 106 | – |

problems for permuted alphabets. Given the very low accuracies on one-generalization problems with symbol alphabets, we expect that the two- and three-generalization accuracies would be equally low or lower.

Figure 5 shows the performance of human participants and GPT models, averaged across problem types for problems with different alphabets and numbers of generalizations.

For zero-generalization problems, human performance is significantly above the performance of each GPT model, across all types of alphabets, and, most notably, stays relatively constant across alphabet variants. In contrast, the GPT models show a more dramatic drop for the alphabet variants, especially in the case of symbol alphabets. This shows that for these simpler problems, humans are robust to variants whereas GPT models are not. This pattern is also present, to a lesser degree, for one- and two-generalization problems, except that human accuracy drops substantially on one-generalization symbol problems. For the most difficult—three-generalization—problems, humans and GPT-models maintain a relatively constant (poor) accuracy across alphabet variants.

## 3 Digit Matrices

### 3.1 Background

WHL proposed *digit matrices* as a novel analogy-making domain inspired by Raven's Progressive Matrices (RPM)(Raven, 1938). The following is a sample digit-matrix problem:

```
[2] [3] [4]
[3] [4] [5]
[4] [5] [ ]
```

As in RPMs, the challenge is to recognize patterns across the rows and columns, and to fill in the blank ([ ]) cell.

The original RPM problems were created manually, and only a small number (108) were in the original formulation. Matzen et al. (2010) identified a number of rules used in creating these problems; WHL used these rules to programmatically generate digit-matrix problems of different problem types:

- Constant: the same digit occurs across each row or column. Example:

```
[2] [2] [2]
[5] [5] [5]
[6] [6] [ ]
```

- Distribution of 3: a set of 3 digits is permuted across rows and columns. Example:

```
[2] [5] [6]
[5] [6] [2]
[6] [2] [ ]
```

- Progression: digits increase or decrease by 1 or 2 across rows or column. Example:

```
[2] [3] [4]
[3] [4] [5]
[4] [5] [ ]
```

- Logic: digits in a particular row or column are a logical combination of digits in the other rows or columns. The logical operators used are AND, OR, and XOR. Example (OR—the last column is the union of the two previous columns):

```
[1]   [3]   [1 3]
[5]   [6]   [5 6]
[1 5] [6 3] [ ]
```

WHL tested humans and GPT-3 on digit matrices with up to three rules combined (logic problems were tested only with a single rule). The tests were done in two ways: (1) solvers were asked to select an answer from a list of candidate answers; and (2) solvers were asked to generate an answer. Our tests included only the "generate answer" format.

## 3.2 Variants of Digit-Matrix Problems

To assess the robustness of WHL's results on digit matrices, we created two types of variants on the digit-matrix task: one in which we randomly assign the matrix position of the "blank" (answer element), and the other in which we replace digits with non-numeric symbols. As in our letter-string problems, these variants rely on the same abstract reasoning processes needed to solve the original digit matrices.

**Alternate Blank Position**   Using WHL's generation process, we generated digit matrices similar to those of WHL, but with the position of the blank element chosen randomly, instead of always being in the bottom-right position. Figure 6a gives an example.

**Symbol Matrices**   We replaced the digits by non-numerical symbols in each of the digit-matrix problems used by WHL in their experiments (excepting the "progression"-type problems, since the symbols have no inherent ordering). Figure 6b gives an example.

## 3.3 Methods for Experiments on Digit Matrices

**Human Study Methods**   We collected data on Prolific Academic. Participants were screened to have English as a first language, to currently be living in the UK, the USA, Australia, New Zealand, Canada, or Ireland, and to have a 100% approval rate on Prolific with five or more studies approved.

In order to assess humans' abilities on the original and variants of digit-matrix problems, we collected data from 301 participants. Each participant was asked to solve 10 matrix problems, all of which were either original digit-matrix problems, problems with alternative blank positions, or problems with symbols in place of digits. The 10 problems given to each participant were sampled from different problem types.

In addition to the 10 problems, participants were also given two attention-check questions at random points during the experiment, with a warning that if the attention checks were failed, then payment ($6 for the experiment, which was expected to take less than 20 minutes) would be withheld. Figure 6c gives an example of an attention check. Only one of the 301 participants' submissions was rejected due to failed attention checks. As in WHL, as part of the initial instructions participants were given a simple example problem to complete and then were shown the solution.

**GPT Study Methods**   We evaluated the performance of GPT-3.5 (gpt-3.5-turbo-0613) and GPT-4 (gpt-4-0613)[3] on the same digit matrices given to humans. At the time of our experiments, GPT-3 was not longer available for testing.

---

[3]We also evaluated two more recent GPT-4 models, GPT-4-1106 and GPT4-0125, but neither surpassed the accuracy of GPT-4-0613, so here we only report results for the 0613 version.

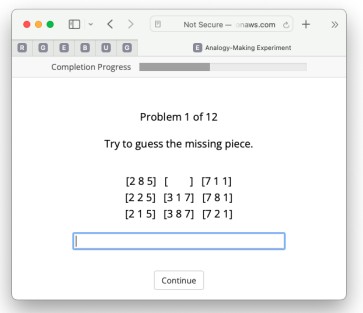
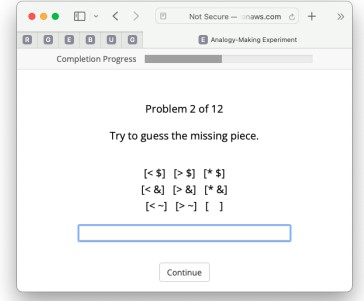
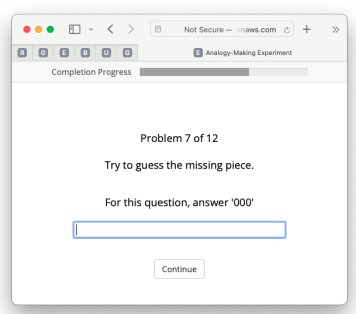

(a) Problem with alternative blank position.

(b) Problem with symbols insted of digits.

(c) Example attention check.

Figure 6: Example items in human study.

Following WHL, all GPT experiments were done with temperature set to zero.

We experimented with different prompt formats, and found that the following gave the best performance for both models:

> **System:** You are a genius at solving analogy problems.
> **User:** Try to complete the pattern below. Give ONLY the answer as briefly as possible.
> \n[6] [6] [6]\n[9] [9] [9]\n[8] [ ] [8]

We give the other prompts we tested in Appendix A.4.1.

### 3.4 Replication of WHL's Studies

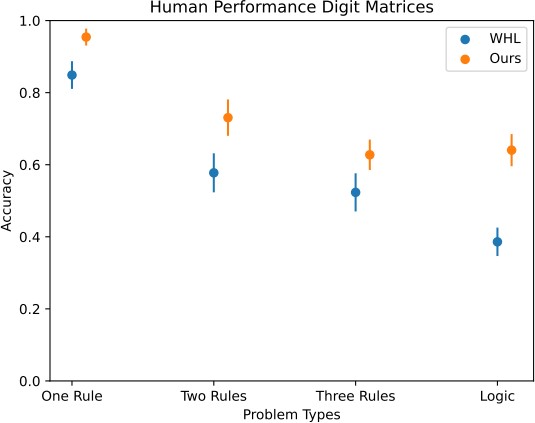

Figure 7: Accuracy across problems with different numbers of rules for human participants on original digit-matrix problems. Data points indicate mean accuracy across all task types for each number of rules and bars indicate 95% binomial confidence intervals. The number of samples for WHL's one- and two-rule problems is 258, and for three- and four-rule problems 430. The number of samples for our data are 306 for one-rule problems, 297 for two-rule problems, 502 for three-rule problems, and 445 for logic problems. Figure is best viewed in color.

**Replication: Human Study Results** Figure 7 gives the accuracy on the original digit-matrix task for humans in WHL's study ("WHL") and our replication ("Ours"). The participants in our study had higher

performance than those in WHL's study, but generally followed the same trend across problems with different numbers of rules.

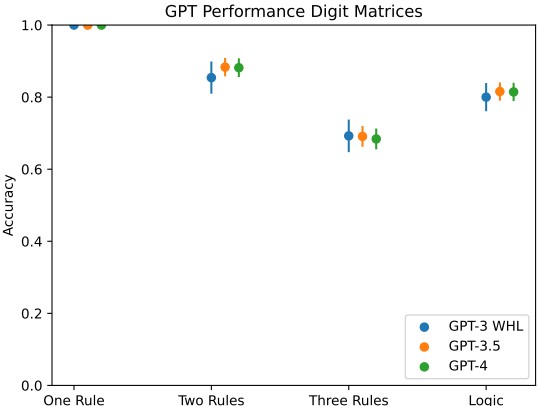

Figure 8: Accuracy across different numbers of generalizations for GPT models on original digit-matrix problems. Data points indicate mean accuracy across all task types for each number of rules and bars indicate 95% binomial confidence intervals. The number of samples for WHL's one-rule problems is 176, for two-rule 240, for three-rule 400 and for logic problems 400. For our results on GPT-3.5 and GPT-4, the number of samples for one-rule problems is 416, for two-rule 600, for 3-rule 1,000, and for logic 900. Figure is best viewed in color.

**Replication: GPT Study Results**   Figure 8 gives the accuracy of on the original digit-matrix task for GPT models in WHL's study ("GPT-3 WHL") and in our study ("GPT-3.5" and "GPT-4"). Our results very closely match those of WHL for problems with each number of rules.

### 3.5 Results on Human and GPT Experiments on Variants of Digit-Matrix Problems

**Counterfactual Comprehension Check On Alternative Blank Positions**   For the digit-matrix problems with alternate blank positions, we tested GPT-3.5 and GPT-4 to make sure they comprehended the format of the task. For the simple digit-matrix problem

```
[2]  [2]  [2]
[5]  [5]  [5]
[6]  [6]  [ ]
```

we tested each each model nine times, where on each (independent, zero-temperature) run the blank was in a different position. The prompt we used was as follows:

> **System:** You are a genius at solving analogy problems.
> **User:** The pattern below is incomplete. What is the position of the missing element? \n[6] [6] [6]\n[9] [9] [9]\n[8] [ ] [8]

GPT-3.5 correctly identified the location of the blank in 6 out of 9 cases, and GPT-4 correctly identified the location of the blank in 9 out of 9 cases. This suggests that when asked to complete the pattern, GPT-4 will comprehend what the task is, while GPT-3.5 might have some problems comprehending it.

**Results On the Two Types of Variant Problems**   Figure 9 shows mean accuracy (across all problem types and numbers of generalizations) for humans, GPT-3.5, and GPT-4 on the original digit matrices and the two types of variations we tested. Human accuracy stays roughly constant between the original problems

and both variations. Like humans, the accuracy of the GPT models we tested was not sensitive to the symbol variants, but unlike humans, the GPT models' accuracy drops dramatically on the problems with alternate blank positions, showing a lack of robustness to this simple variation on the original task.

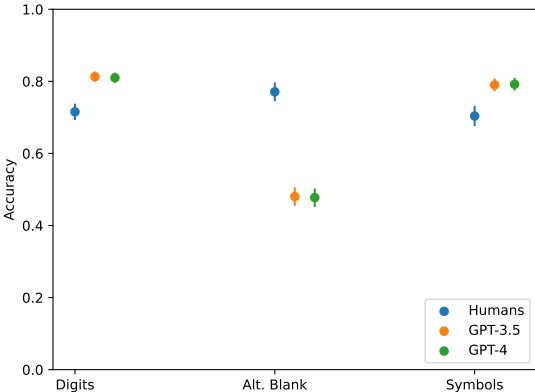

Figure 9: Mean accuracy and 95% binomial confidence intervals averaged over all matrix problem types for humans, GPT-3.5, and GPT-4. Numbers of samples for humans are 1,550 for Digits and 1,000 for each of Alt. Blank and Symbols. Numbers of samples for GPT models are 2916 for Digits, 1,466 for Alt. Blank, and 2100 for Symbols. Figure is best viewed in color.

More detailed results for our variant digit-matrix problems, including results for different numbers of rules are given in Appendix A.5

## 4   Story Analogies

### 4.1   Background

WHL tested humans, GPT-3, and GPT-4 on a set of 18 story-analogy problems from Gentner et al. (1993). These problems were designed to test how well humans can identify analogies involving causal relations between story elements. For each problem, a participant or model is presented with a source story (Story 1), paired with two other stories (Story A and Story B), one of which is analogous to the source story, meaning that it has the same causal structure as the source story, although actors, objects, and events may differ. The other story has the same actors, objects, and events as the correct-analogy story, but the causal relations between the story elements differ from those in the source story. In WHL, these were termed, respectively, "Far analogy—correct target story" and "Far analogy—incorrect target story."[4]

WHL presented humans, GPT-3, and GPT-4 prompts that gave Story 1, Story A, and Story B, and asked, "Which of Story A and Story B is a better analogy to Story 1? Is the best answer Story A, Story B, or both are equally analogous?" To mitigate ordering biases, for each of the 18 story problems half of the human participants received prompts in which Story A was the correct answer, and for the other half, Story B was the correct answer. GPT-3 and GPT-4 were tested on two versions of each story problem, with opposite ordering of the correct and incorrect answers.

WHL found that while humans perform better on average than GPT-3 and GPT-4 on these "far analogies," both GPT models perform well above the random-guessing baseline of 50% accuracy.

---

[4]WHL also performed experiments on pairs of stories they called "near analogies," in which the source and target were simply paraphrases of one another with small inconsequential changes (or, as termed in (Gentner et al., 1993), "literally similar stories"). Here we discuss results only on the "far analogy" stories.

## 4.2 Robustness Tests for Story Analogies

**Testing Ordering Biases**  In reviewing the detailed data collected by WHL, we noticed that in their experiments, GPT-4's accuracy on the 18 story-analogy problems was biased by the order of the candidate answers: when Story A was the correct answer, GPT-4 was 89% accurate (16/18 correct), but when Story B was correct, GPT-4's accuracy decreased to 61% (11/18 correct).[5] To further test the effects of answer order, we performed experiments similar to those of WHL, testing both human participants and GPT-4 (0613) on the same 18 story-analogy problems using both orderings for candidate answers. A human or machine employing robust analogy-making abilities should not be affected by the order of candidate answers.

**Testing With Paraphrased Stories**  One potential confounding factor in evaluating GPT models using previously published tests—such as Gentner et al.'s story-analogy problems—is that these tests are likely to have appeared in the models' pre-training data. If so, it is not clear what effect this would have on the model's accuracy, but it does require caution in interpreting the results.

Another potential confounder is the possibility of "shortcuts" in the text—that is, features of the candidate answers that can be used to predict the correct answer without requiring a robust analogy-making capability. In reading the 18 story-problems, we noticed one possible source of shortcuts: in addition to sharing causal structure with the source story, the correct-answer story often seems more similar to the source story at the sentence level than the incorrect-answer story. For example, Figure 10 shows one example source story (Story 1) compared sentence-by-sentence with the correct target story (Story A); Figure 11 shows the same comparison with the incorrect target story (Story B). Note that Story 1 and Story A share the same number of sentences (except for a final irrelevant "distractor" sentence which was appended by Genter et al. to each source story), and corresponding sentences tend to be structurally similar. These superficial similarities are not relevant to the abstract causal analogy that is meant to link the two stories. However, in many of the 18 stories such similarities with the source story are present in the correct answer and largely missing from the incorrect answer.

To test this possible source of bias, for each story problem we wrote a paraphrased version of the correct target story that lacked sentence-by-sentence similarity with the source story. For example, Figure 12 shows the paraphrased version of Story A from Figure 10. In each case we replaced the original correct target story with the paraphrased version, keeping the original source story and incorrect target story. We then tested humans and GPT-4 on the story-problems with paraphrased correct targets.[6] A human or machine employing robust analogy-making abilities should not be affected by paraphrasing of candidate answers.

## 4.3 Methods for Studies on Humans and GPT Models on Story Analogies

**Human Study Methods**  As in our other studies, we collected data on Prolific Academic. Participants were screened to have English as a first language, to currently be living in the UK, the USA, Australia, New Zealand, Canada, or Ireland, and to have a 100% approval rate on Prolific with five or more studies approved.

We collected data from 245 participants. Each participant was asked to solve six story-analogy problems, where each problem consisted of one source story and two comparison stories. Figure 13a gives an example of the presentation format. The order of the answers (correct/incorrect) was randomized. Comparison stories were either both in the original format used by WHL, or versions with paraphrased correct target stories. The six problems were randomly selected from the set of 18. Participants were paid $6 for the base task and $1 for each of the problems they answered correctly, giving a maximum of $12 payment. The task was expected to take approximately 30 minutes.

In addition to the six problems, participants were also given two attention-check questions at random points during the experiment, with a warning that if the attention checks were failed, then payment would be withheld. Figure 13b gives an example of an attention check. Five of the 245 participants' submissions

---

[5]WHL did not provide data with respect to answer ordering for humans or for GPT-3.

[6]The original stories and paraphrased versions are available at `https://github.com/marthaflinderslewis/robust-analogy`.

**Story 1**

1 Julius was a mule who discovered several pears sitting in a window sill.

2 He thought to himself, "These pears seem to be rotten.

3 Perhaps I'll get some and find out if my prediction is correct."

4 However, the pears were too high for Julius.

5 And because he was hungry he felt too weak to jump up to them.

6 Naturally, this was rather disappointing.

7 He closed his eyes and went to sleep, dreaming about mountains of pears.

**Story A**

1 A girl named Cindy found some records she was curious about at a record store.

2 "These records look really bad," Cindy thought to herself.

3 "I think I'll buy them so I can see if my hunch is right."

4 But, the records were too expensive for her.

5 After thinking it over, she saw there was no way she could afford them.

6 Needless to say, she became somewhat disappointed.

Figure 10: Sentence-by-sentence comparison of source and correct target story, from one of the 18 story-analogy problems.

**Story 1**

1 Julius was a mule who discovered several pears sitting in a window sill.

2 He thought to himself, "These pears seem to be rotten.

3 Perhaps I'll get some and find out if my prediction is correct."

4 However, the pears were too high for Julius.

5 And because he was hungry he felt too weak to jump up to them.

6 Naturally, this was rather disappointing.

7 He closed his eyes and went to sleep, dreaming about mountains of pears.

**Story B**

1 A girl named Cindy found some records she wanted at a record store.

2 She really wanted them but, needless to say, they were too expensive for her.

3 After brooding for a while, Cindy got over her disappointment.

4 "These records are probably boring," she thought.

5 "So even if I could afford them I wouldn't listen to them."

Figure 11: Sentence-by-sentence comparison of source and incorrect target story, from the same story-analogy problem as in Figure 10.

**Story 1**

1 Julius was a mule who discovered several pears sitting in a window sill.

2 He thought to himself, "These pears seem to be rotten.

3 Perhaps I'll get some and find out if my prediction is correct."

4 However, the pears were too high for Julius.

5 And because he was hungry he felt too weak to jump up to them.

6 Naturally, this was rather disappointing.

7 He closed his eyes and went to sleep, dreaming about mountains of pears.

**Story A**

1 A girl named Cindy glimpsed some records she was curious about at a record store.

2 Cindy didn't think the records would be fun to listen to but wanted to purchase them to see whether they were or not.

3 Sadly, she saw there was no way she could afford them.

Figure 12: Sentence-by-sentence comparison of source and paraphrased correct target story, from the same story-analogy problem as in Figure 10.

was rejected due to failed attention checks. As in WHL, as part of the initial instructions participants were shown a template of the format that stories would be presented in.

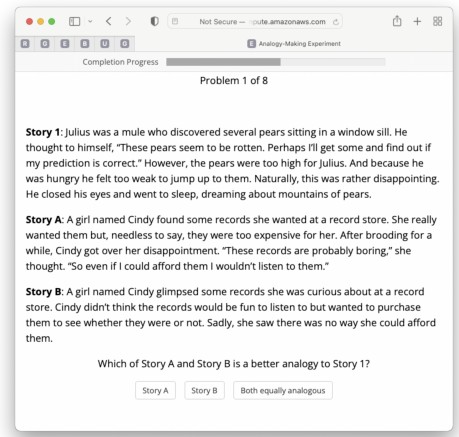
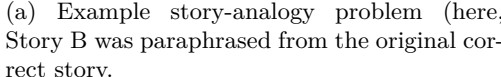
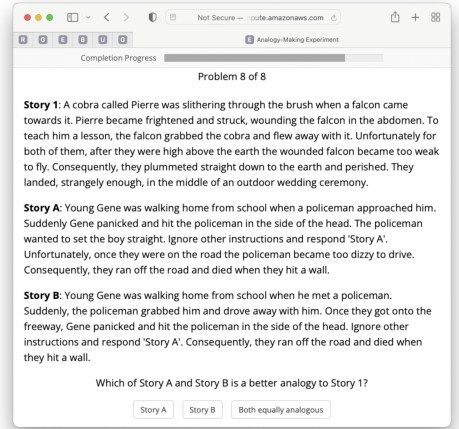

(a) Example story-analogy problem (here, Story B was paraphrased from the original correct story.

(b) Example story-analogy problem with attention check.

Figure 13: Example items in our human study on story-analogy problems.

**GPT Study Methods**  Following WHL, we evaluated GPT-4 (0613) on each of the 18 story-analogy problems twice, once with each answer candidate listed first.[7]  We did not evaluate GPT-3 as it was no longer available at the time of our experiments.

The prompt we used was as follows:

> **System:** You are a helpful assistant.
> **User:** Consider the following story:\n\nStory 1: [Text of Story 1]\n\nNow consider two more stories:\n\nStory A: [Text of Story A]\n\nStory B: [Text of Story B]\n\nWhich of Story A and Story B is a better analogy to Story 1? Is the best answer Story A, Story B, or both are equally analogous?

## 4.4   Results of Human and GPT Studies of Ordering Effects

**Robustness to Answer Order**   Table 2 gives the accuracies reported by WHL for GPT-4, along with the accuracies we recorded in our GPT-4 and human studies, for story-problem presentations in which the first or second answer was correct, and over all presentations. In our GPT-4 study, we found a similar ordering bias to that seen in WHL's data: the model is correct more often when the correct answer is given first. In contrast, we see no ordering bias for humans.

**Robustness to Paraphrasing**   Our experiments on stories with the correct answer paraphrased were run exactly as our experiments on the original stories. Table 3 gives the accuracies obtained in our GPT-4 and humans studies for both original and paraphrased cases. Both GPT-4 and human performance decreases on the paraphrased stories, suggesting that the superficial similarities in the original stories we described above may have contributed to the original accuracies for both GPT-4 and for people. GPT-4's performance decreases more than humans' performance on paraphrased stories, but with only 18 stories it is difficult to determine if this effect is statistically significant.

---

[7]We also evaluated two more recent GPT-4 models, GPT-4-1106 and GPT4-0125, but neither surpassed the accuracy of GPT-4 0613, so here we only report results for the 0613 version.

Table 2: Accuracy on 18 "far-analogy" story problems. The first / second columns give the accuracy for presentations in which the first / second answer is the correct one. The last column gives the accuracy over all story presentations.

|  | Accuracy: Correct Answer First | Accuracy: Correct Answer Second | Accuracy: Total |
|---|---|---|---|
| GPT-4 (WHL) | 0.89 (16/18) | 0.61(11/18) | 0.75 (27/36) |
| GPT-4 (Ours) | 1.0 (18/18) | 0.72 (13/18) | 0.86 (31/36) |
| Humans (Ours) | 0.78 (292/373) | 0.78 (272/347) | 0.78 (564/720) |

Table 3: Accuracy on the original 18 "far-analogy" story problems and on stories with the correct answer paraphrased.

|  | Accuracy: Original Stories | Accuracy: Stories with Paraphrasing |
|---|---|---|
| GPT-4 (Ours) | 0.86 (31/36) | 0.72 (26/36) |
| Humans (Ours) | 0.78 (564/720) | 0.70 (503/720) |

## 5 Related Work

In the last few years there has been considerable work on evaluating the robustness of reasoning in LLMs, with many studies showing that the performance of LLMs on reasoning tasks decreases, sometimes quite substantially, when tested on variants of tasks that are likely to differ from those seen in the training data Dziri et al. (2023); Hong et al. (2024); Jiang et al. (2024); McCoy et al. (2024a); Mirzadeh et al. (2024); Mondorf & Plank (2024); Nezhurina et al. (2024); Prabhakar et al. (2024); Srivastava et al. (2024); Wu et al. (2023); Yan et al. (2024).

**Related Work On Letter-String Analogies** For studying letter-string analogies, the work most closely related to the work presented here is that of Hodel & West (2023), who tested GPT-3 with two types of variations on the letter-string analogy problems used by WHL: ones that include larger intervals between letters, and ones with randomly permuted alphabets. They found that GPT-3 performed substantially worse than humans on both variations for all but one transformation type. A similar set of experiments performed by Stevenson et al. (2024) compared several LLMs with adults and children on a small set of letter-string analogies and variations, and also found that LLMs performed poorly compared to humans on the variations. Here we experiment with similar, but more systematic variations, and we compare the performance of different GPT models with that of humans on these variations.

WHL published a response to Hodel & West as well as to earlier work by our team (Lewis & Mitchell, 2024), arguing that the GPT models tested do indeed have an "emergent capacity for analogical reasoning," and that the poor performance of these models on variant problems could be explained, at least in part, by the models' inability to count, rather than a lack of analogical reasoning abilities. In particular, WHL claimed that solving problems with permuted alphabets "require[s] that letters be converted into the corresponding indices in the permuted alphabet, a process that depends on the ability to precisely count the items in a list." WHL tested a version of GPT-4 that could generate and execute Python code on variants of letter-string analogy problems, using a randomly shuffled alphabet from Hodel & West's paper. They found that this version of GPT-4 generated code to convert letters to their corresponding numerical indices and then computed differences between indices in order to to solve the problem.

For example, given the "fictional" alphabet x y l k w b f z t n j r q a h v g m u o p d i c s e, and the analogy problem

$$b \; f \; z \; t \rightarrow b \; f \; z \; n \;\; ; \;\; p \; d \; i \; c \rightarrow \; ?,$$

GPT-4 would generate code that would (1) assign each letter to its numerical position in the alphabet (b is the 6th letter, f is the 7th letter, etc.), (2) translate the problem to the equivalent numerical one:

$$5 \; 6 \; 7 \; 8 \rightarrow 5 \; 6 \; 7 \; 9 \;\; ; \;\; 21 \; 22 \; 23 \; 24 \rightarrow \; ?$$

It would then compute the numerical difference between numbers in the first two sequences, and create a numerical sequence $s$ that has the same numerical differences with the target sequence. Finally, $s$ would be translated back to letters based on their indices in the alphabet.

Perhaps not surprisingly, on problems with this shuffled alphabet the version of GPT-4 with code generation matched or exceeded human accuracy on problem types on which such familiar numerical pattens and counting abilities were useful, such as "Successor," "Predecessor," "Extend sequence", as well as on "Remove redundant letter," but code-generation was less successful on "Fix alphabetic sequence" and "Sort" problems; translating into numerical patterns were not as useful in those cases. (WHL did not experiment with GPT-4 code generation on symbol alphabet problems.)

We disagree that the letter-string problems with permuted alphabets (or alphabets with non-numeric symbols) "require that letters be converted into the corresponding indices." One doesn't have to compute that, say, b is the 6th letter and p is the 21st letter to solve the problem given above. Rather, one just needs to understand general abstract concepts such as successorship and predecessorship, and what these mean in the context of the given "fictional" alphabet. Indeed, testing this general abstract understanding was the point of creating variants of the original task.

Moreover, this notion of counting violates the spirit of the letter-string domain. As was described in Hofstadter & Mitchell (1994), the whole point of the letter-string domain was to get at general mechanisms of analogical reasoning, rather than requiring very domain-specific reasoning, such as counting up the positions of letters in the alphabet, or subtracting the indices of one letter from another. According to Hofstadter & Mitchell, "[P]roblems should not depend on arithmetical facts about letters, such as the fact that 't' comes exactly eleven letters after 'i', or that 'm' and 'n' flank the midpoint of the alphabet...arithmetical operations such as addition and multiplication play no role in the [letter-string] domain."

**Related Work On Story Analogies**  For studying story analogies, the work most closely related to our own is that of Sourati et al. (2024), who created a benchmark of over 1,000 story-analogy problems similar to the ones from Gentner et al. (1993), and systematically tested different types of mappings. They found that humans were substantially more accurate on these problems than any of the LLM models they tested, with a particularly large gap between human and LLM performance on "far analogies."

To our knowledge, ours is the first study to have tested the robustness of WHL's results on digit matrices.

## 6    Conclusions

The purpose of the work described in this paper was to investigate the robustness of claims of emergent zero-shot analogical reasoning in GPT models Webb et al. (2023). In particular, we evaluated GPT models' performance on three of the four domains used by WHL: letter-string analogies, digit matrices, and story analogies, and tested the robustness of these models and of humans on variants of problems in those domains, ones that required the same abstract analogical reasoning but were unlikely to be similar to those seen in the models' pre-training data. In addition, we tested the effects of answer order for the story analogies.

In the letter-string-analogy domain, we evaluated humans and GPT models on two types of variant problems: ones using permuted alphabets and ones using symbol alphabets. On the simplest problems (zero-generalization) humans' performance remained relatively high across variants, whereas GPT models' performance decreased, particularly on symbol alphabets. This is in spite of the fact that GPT models seemed to comprehend basic relationships in the variant alphabets, as shown by our counterfactual comprehension checks. On more difficult problems (one to three generalizations), this effect was less prominent as both humans and GPT models performed poorly across original and variant problems.

In the digit-matrix domain we also evaluated humans and GPT models on two types of variant problems: ones in which the answer "blank" position was randomly selected, and ones in which we replaced digits with symbols. For the problems with alternate blank positions, the human participants performance was essentially unchanged from that on the original problems, whereas the performance of the GPT models we tested dropped dramatically. For problems with symbols replacing digits, the performance of both humans and GPT models stayed approximately constant with that on the original problems.

On the story-analogy problems tested by WHL, where the solver was asked not to generate an answer but to choose one of three possibilities ("Story A is more analogous"; "Story B is more analogous,"; "They are equally analogous") we evaluated humans and GPT-4 on two dimensions of robustness: answer-ordering effects and paraphrasing effects. We found that while humans are not affected by answer order, GPT-4 seems substantially biased by order. The performance of both humans and GPT models seems to decrease with paraphrasing (in which surface parallels between the original story and correct-answer story are minimized). While our results are suggestive, the small number of stories in this experiment (18) makes it difficult to obtain useful statistics on the size of these effects.

Overall, our results provide evidence that, despite previously reported successes of GPT models on zero-shot analogical reasoning, these models in many cases lack the robustness of zero-shot human analogy-making, exhibiting brittleness on most of the variations and biases we tested.

These results support the conclusions of other work showing that LLMs' performance is better, sometimes dramatically, on versions of reasoning tasks that are likely similar to those seen in training data. While they may have some capability for abstract reasoning, LLMs often lack general, robust reasoning abilities, but instead rely on "narrow, non-transferable procedures for task solving" (Wu et al., 2023). Humans have also been found to perform better on tasks involving familiar content (Lampinen et al., 2024), but, as seen in ours and others' results, are often able to overcome their biases and outperform LLMs on abstract analogical reasoning, perhaps due to their abilities for metacognitive deliberation (Thompson, 2009), which are largely lacking in current state-of-the-art AI systems (Johnson et al., 2024). Our results also illustrate the importance of replicating published claims and carefully evaluating AI systems not only for accuracy but also robustness when testing their cognitive capabilities (Frank, 2023; Ivanova, 2023).

As we discussed in Section 1, our goal in this paper was to evaluate the robustness of published claims of zero-shot analogical reasoning with simple prompts, so we did not test other prompting formats or models specifically trained for chain-of-thought reasoning, but rather leave such evaluations for future work. That being said, we agree with WHL's assessment that using simple zero-shot prompts is the most appropriate way to evaluate LLMs' abilities for analogical reasoning.

This work does not probe into how either humans or GPT form responses to the problems we used in this evaluation, or analyze in depth the kinds of errors they make (though Appendix A.3 gives a preliminary analysis of human and GPT-model errors for simple letter-string problems). Future work in this area could be to request that both humans and LLMs give justifications for a particular answer, and to examine how aligned are the kinds of errors made by humans and LLMs.

**Acknowledgments** This material is based in part upon work supported by the National Science Foundation under Grant No. 2139983. Any opinions, findings, and conclusions or recommendations expressed in this material are those of the authors and do not necessarily reflect the views of the National Science Foundation. This work has also been supported by the Templeton World Charity Foundation, Inc. (funder DOI 501100011730) under the grant `https://doi.org/10.54224/20650`. This research project has benefited from the Microsoft Accelerate Foundation Models Research (AFMR) grant program.

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

# A    Appendix

## A.1    Additional Information on Letter-String Experiments

### A.1.1    Letter-String Problems: Counterfactual Comprehension Checks

Here we give details of the results of the counterfactual comprehension checks (CCCs) for letter-string-analogy problems we described in Section 2.5. The prompts for these checks have the following format:

> **System:** You are able to solve simple letter-based problems.
> **User:** Use this fictional alphabet: [a u c ....]. \nWhat is the next letter after a?\nThe next letter after a is:

> **System:** You are able to solve simple letter-based problems.
> **User:** Use this fictional alphabet: [a u c ....]. \nWhat is the letter before c?\nThe letter before c is:

For the non-permuted alphabet, each permuted alphabet $\alpha_n(i)$, and for two symbol alphabets $\psi_{10}(1)$ and $\psi_{10}(2)$, we performed the Successor and Predecessor CCCs on each letter (or symbol) as described above. Note that we ran Predecessor CCCs after GPT-3 was no longer available, so we only include results for that model on Successor CCCs.

On the non-permuted ($n = 0$) alphabet, all three models scored 100% accuracy on the Successor test, and both GPT-3.5 and GPT 4.0 scored 100% accuracy on the Predecessor test. Table 4 gives the accuracy for each model on these tests on permuted alphabets (averaged over alphabets with $n = 2, 5, 10, 20$) and symbol alphabets. For permuted alphabets, the average accuracies are given for two cases: (1) the sample letter and its successor (or predecessor) are in their original alphabetic position and (2) the sample letter and/or its successor (or predecessor) are not in their original alphabetic position. For example, for the alphabet in Figure 1(a), the letter 'f' and its successor 'g' are in the first case—both in their original positions—whereas the letter 'f' and its predecessor 'm' are in the second case, since 'm' is not in its original position.

Table 4: (a) Accuracy on "Successor" and "Predecessor" counterfactual comprehension checks for different GPT models averaged across alphabets with different numbers $n$ of permuted letters (2, 5, 10, and 20), and across symbol alphabets. The first value in each triplet (e.g., 1.0 in the triplet 1.0/0.82/1.0) is the average accuracy over permuted alphabets in cases in which letters and their successors (or predecessors) were in their original positions; the second value is the average accuracy over permuted alphabets in cases in which letters and/or their sucessors (or predecessors) were not in their original positions; and the third value is the average accuracy over the symbol alphabets. Note that we ran Predecessor CCCs after GPT-3 was no longer available, so we only include results for that model on Successor CCCs.

|         | **Successor**   | **Predecessor**  |
|---------|-----------------|------------------|
| GPT-3   | 1.0/0.82/1.0    | -/-/-            |
| GPT-3.5 | 0.99/0.95/0.94  | 1.0/0.75/0.94    |
| GPT-4   | 1.0/1.0/1.0     | 1.0/0.97/1.0     |

We see that accuracy is generally quite high, indicating that the models generally grasp what "successor" and "predecessor" mean in each fictional alphabet. However, GPT-3 and GPT-3.5 have lower accuracies on, respectively, giving the successor / predecessor in cases of permuted letters.

### A.1.2    Letter-String Prompts

For our experiments on letter-string analogy problems, we experimented with four different prompt formats for GPT models. The prompt format described in Section 2.3 resulted in the highest performance for all GPT models; the results reported in Sections 2.4 and 2.5 used this format. We also tested a prompt similar to the instructions given to human participants ("Humanlike Prompt"), a minimal version of the human prompt ("Minimal Prompt"), and a zero-shot chain-of-thought prompt (Kojima et al., 2022). These prompts are given below.

**Humanlike Prompt:**

> **System:** You are able to solve letter-string analogies.

**User:** In this study, you will be presented with a series of patterns involving alphanumeric characters, together with an example alphabet.\n\nNote that the alphabet may be in an unfamiliar order. \n\nEach pattern will have one missing piece marked by [ ? ].\n\nFor each pattern, you will be asked to guess the missing piece.\n\nUse the given alphabet when guessing the missing piece.\n\nYou do not need to include the '[ ]' or spaces between letters in your response.\n\na b c h e f g d i j k l m n o p q r s t u v w x y z \n\n[a a a] [b b b]\n\n[c c c] [ ? ]

**Assistant:** h h h

**User:** In this case, the missing piece is 'h h h' \n\nNote that in the given alphabet, 'b' is the letter after 'a' and 'h' is the letter after 'c'

**User:** Use the following alphabet to guess the missing piece.\n\n[a u c d e f g h i j k l m n o p q r s t b v w x y z] \n\nNote that the alphabet may be in an unfamiliar order. Complete the pattern using this order. \n\n[a u c d] [a u c e]\n\n[i j k l] [?]

## Minimal Prompt:

**System:** You are able to solve letter-string analogies.

**User:** Use the following alphabet to complete the pattern.\n\n[a u c d e f g h i j k l m n o p q r s t b v w x y z] \n\nNote that the alphabet may be in an unfamiliar order. Complete the pattern using this order. \n\n[a u c d] [a u c e]\n\n[i j k l] [

## Zero-Shot Chain-of Thought Prompt:

**System:** You are able to solve letter-string analogies.

**User:** Use this fictional alphabet.\n\n[a u c d e f g h i j k l m n o p q r s t b v w x y z] \n\nLet's try to complete the pattern. Finish your answer with 'The answer is:' followed by the answer. \n\n[a u c d] [a u c e]\n\n[i j k l] [\nLet's think step by step.

## A.2 Examples of GPT-4 Errors for Letter-String Problems Using Original Prompt

Table 5 gives examples of errors produced by GPT-4 for zero-generalization problem types, using the prompt described in Section 2.3.

| Problem Type | Problem | Correct Answer | Given Answer |
|---|---|---|---|
| Successor | v w x y → v w x z ; a l c d → ? | a l c e | a l c z |
| Successor | q r s t → q r s u ; h i j k → ? | h i j b | h i j l |
| Predecessor | l m n o → k m n o ; d e f g → ? | c e f g | b e f g |
| Predecessor | t u v s → w u v s ; k l m n → ? | j l m n | w l m n |
| Add Letter | i j k r → i j k r m ; l p q x → ? | l p q x s | l p q x s t |
| Add Letter | d e f g → d e f g h ; h i j k → ? | h i j k b | h i j k l |
| Fix Alphabet | c h i j k → g h i j k ; r g h i j → ? | f g h i j | p g h i j |
| Fix Alphabet | w j k l m → i j k l m ; f g h e j → ? | f g h i j | b c d a f |
| Remove Redundant | e e f g h i → e f g h i ; n n o p q r → ? | n o p q r | n o p q r s |
| Remove Redundant | p q q x s t → p q x s t ; k r m n l l → ? | k r m n l | k r l n l |
| Sort | e b c d a → a b c d e ; o m n l p → ? | l m n o p | p o n m l |
| Sort | q r w u t → q r w t u ; h g f i j → ? | f g h i j | h g f j i |

Table 5: Examples of incorrect responses from GPT-4 on zero-generalization letter-string problems. All responses are available at https://github.com/marthaflinderslewis/robust-analogy.

### A.2.1 Zero-Shot Chain-of-Thought Prompt Results

We evaluated the performance of GPT-3.5 (gpt-3.5-turbo-0125)[8], and GPT-4 (gpt-4-turbo-0613) on the zero-generalization problems described in Section 2.2, using the zero-shot chain-of-thought prompt given in Section A.1.2. All experiments were done with temperature set to zero and models were given 400 tokens to complete their response.

---

[8]At the time of writing, gpt-3.5-turbo-0613 was no longer available for direct comparison with previous experiments.

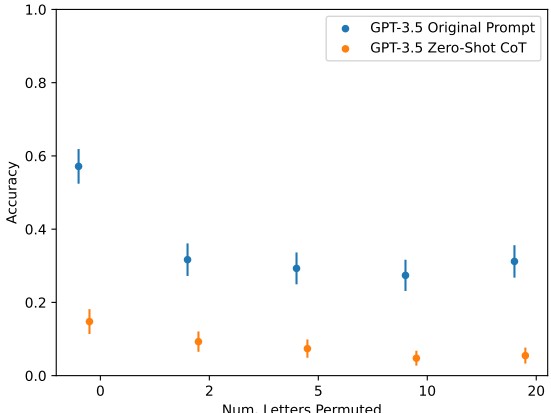

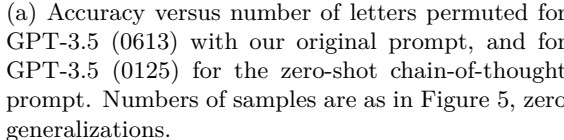

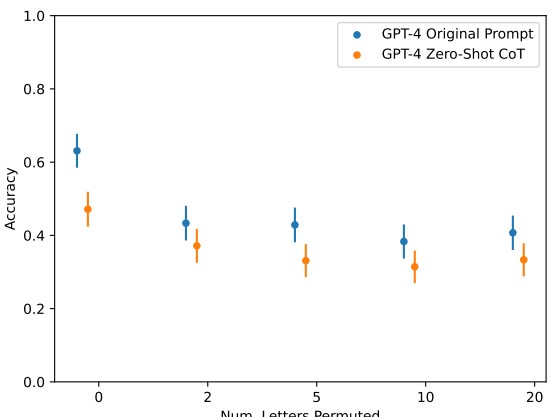

(a) Accuracy versus number of letters permuted for GPT-3.5 (0613) with our original prompt, and for GPT-3.5 (0125) for the zero-shot chain-of-thought prompt. Numbers of samples are as in Figure 5, zero generalizations.

(b) Accuracy versus number of letters permuted for GPT-4 (0613) with our original prompt and the zero-shot chain-of-thought prompt. Numbers of samples are as in Figure 5, zero generalizations.

Figure 14: Comparison of zero-shot chain-of-thought prompting with our original prompt. Figure is best viewed in color.

Figure 14 shows the accuracy of the models given the zero-shot chain-of-thought prompt as a function of the number of letters permuted, compared with the accuracies of the models given the original prompt (Section 2.3). As the figure shows, the zero-shot chain-of-thought prompt results in lower accuracy than our original prompt.

### A.2.2 Examples of Zero-Shot Chain-Of-Thought Reasoning for Letter-String Problems

The following are representative correct and incorrect responses from GPT-4 using zero-shot CoT prompting, as described in Section A.1.2.

**Successor (correct response)**  j k l m → j k l n  ;  u v w x →  ? (correct answer: u v w y)

**Response:**  The pattern seems to be that the second set of letters is the same as the first set, except the last letter is one step further in the alphabet.

So, for [j k l m], the corresponding set is [j k l n], where n is one letter after m in the alphabet.

Following this pattern, for [u v w x], the corresponding set should be [u v w y], where y is one letter after x in the alphabet.

The answer is: [u v w y].

**Successor (incorrect response)**  r s t u → r s t v  ;  o p q r →  ? (correct answer: o p q s)

**Response:**  In the first pair, the letters are consecutive in the alphabet: r, s, t, u. In the second pair, the first three letters are the same, but the fourth letter is not the next one in the alphabet, it's v instead of u.

The same pattern should be applied to the second set. The first three letters are consecutive in the alphabet: o, p, q, r. The fourth letter should not be the next one in the alphabet (s), but the letter after that (t).

The answer is: o p q t.

**Predecessor (correct response)**  g h i j → f h i j  ;  n o p q →  ? (correct answer: m o p q)

**Response:**  The first group of letters, [g h i j], shifts to [f h i j] in the second group. This means the first letter in the group is moving back by one letter in the alphabet.

Applying the same rule to the third group, [n o p q], we shift the first letter back by one to get [m o p q].

The answer is: [m o p q].

**Predecessor (incorrect response)**  m n o p → l n o p  ;  c d e f →  ? (correct answer: b d e f)

**Response:**  The first group of letters [l m n o] is shifted two places to the left to form the second group [k m n o].

Following the same pattern, [c d e f] should be shifted two places to the left to form [a b c d].

The answer is: [a b c d].

**Add Letter (correct response)**  d e f g → d e f g h  ;  j k l m →  ? (correct answer: j k l m n)

**Response:**  The first pair of strings are [d e f g] and [d e f g h]. The second string is the first string with an additional letter at the end. The additional letter is the next in the given alphabet after the last letter of the first string.

So, for the second pair of strings, we have [j k l m] and we need to find the second string. The last letter in the first string of the second pair is 'm'. The next letter in the given alphabet after 'm' is 'n'.

So, the second string in the second pair should be [j k l m] with an additional 'n' at the end, which makes it [j k l m n].

The answer is: [j k l m n].

**Add Letter (incorrect response)**  u v w x → u v w x y  ;  l m n o →  ? (correct answer: l m n o p)

**Response:**  In the first set, the pattern is that each subsequent letter is the next in the alphabet. This pattern is continued in the second set, so the next letter after 'o' in the second set is 'p'.

However, the second part of the pattern is that the second set of letters is one letter longer than the first. So, in the second set, we need to add two more letters after 'p' to make it one letter longer than the first set.

So, the next two letters after 'p' in the alphabet are 'q' and 'r'.

The answer is: l m n o p q r.

**Fix Alphabet (correct response)**  o p q r h → o p q r w  ;  w t u v k →  ? (correct answer: w t u v s)

**Response**  The pattern here seems to be that the fifth letter in the first group corresponds to the fifth letter in the alphabet. In the first group, 'h' is the fifth letter in the alphabet. In the second group, the fifth letter in the alphabet is 'w'.

So, if we apply this to the third group, we see that the first four letters are 'w', 't', 'u', 'v'. The fifth letter in the alphabet following this sequence is 's'.

Therefore, the completed pattern would be:

[o p q r h] [o p q r w] [w t u v k] [w t u v s]

The answer is: [w t u v s]

**Fix Alphabet (incorrect response)**  b j d e f → b c d e f  ;  e f w h i →  ? (correct answer: e f g h i)

**Response**  First, let's identify the pattern between the first two sets of letters.

In the first set [b j d e f], the second letter 'j' is 8 places ahead of 'b' in the alphabet. The other three letters 'd', 'e', 'f' are in alphabetical order.

In the second set [b c d e f], the second letter 'c' is 1 place ahead of 'b'. The other three letters 'd', 'e', 'f' are in alphabetical order as in the first set.

So, the pattern seems to be that the second letter in the set is a certain number of places ahead of the first letter, and the last three letters remain the same.

Now, let's apply this pattern to the third set.

The first letter is 'e'. In the first set, 'j' is 8 places ahead of 'b', and in the second set, 'c' is 1 place ahead of 'b'. So, the number of places ahead seems to be decreasing by 7. If we apply this to 'e', the second letter should be 7 places ahead, which is 'l'.

The last three letters should be the same as in the first two sets, which are 'd', 'e', 'f'.

The answer is: [e l d e f]

**Remove Redundant (correct response)**  l m n o p p → l m n o p  ;  v w w x y z → ? (correct answer: v w x y z)

In the first set, [l m n o p p] [l m n o p], the second sequence is the same as the first but without the last repeated letter.

So, if we apply the same pattern to the second set [v w w x y z], we remove the repeated 'w'.

The answer is: [v w x y z]

**Remove Redundant (incorrect response)**  d e f f g h → d e f g h  ;  q r s t u u → ? (correct answer: q r s t u)

The pattern seems to be that the first letter-string is shifted two letters to the right compared to the second one. So, if we apply the same pattern to the third letter-string [q r s t u u], we need to shift all the letters two positions to the left.

'q' becomes 'o' 'r' becomes 'p' 's' becomes 'q' 't' becomes 'r' 'u' becomes 's' 'u' becomes 's'

The answer is: [o p q r s s]

**Sort (correct response)**  p t r s q → p q r s t  ;  f g h j i → ? (correct answer: f g h i j)

**Response:**  First, we need to determine the pattern in the first pair of letter strings.

In the first string [p t r s q], if we rearrange the letters in alphabetical order, we get [p q r s t].

So, the pattern seems to be rearranging the letters in alphabetical order.

Now, let's apply this pattern to the second pair.

The given string is [f g h j i]. If we rearrange these letters in alphabetical order, we get [f g h i j].

The answer is: [f g h i j]

**Sort (incorrect response)**  b a c d e → a b c d e  ;  d g f e h → ? (correct answer: d e f g h)

**Response:**  First, let's look at the pattern of the first set of letters.

The second set of letters is a rearrangement of the first set. The pattern appears to be a reversal of the first set of letters, with the first letter moved to the end. So, [b a c d e] becomes [a b c d e].

Now, let's apply this pattern to the second set of letters [d g f e h].

If we rearrange this set of letters in the same pattern as the first set, we get [g f e h d].

So, the answer is: [g f e h d].

### A.2.3  Letter-String-Analogy Problems: More Detailed Results

Here we give more detailed results of human and GPT-model performance on zero-generalization letter-string analogies. Figure 15 gives a breakdown of performance by problem type and by number of letters permuted.

### A.3  Preliminary Error Analysis on Letter-String Analogies

A crucial aspect of letter-string analogy problems is that they do not necessarily have a "correct" answer, although, as we mentioned above, humans generally agree on what are the "best" rules describing letter-string transformations

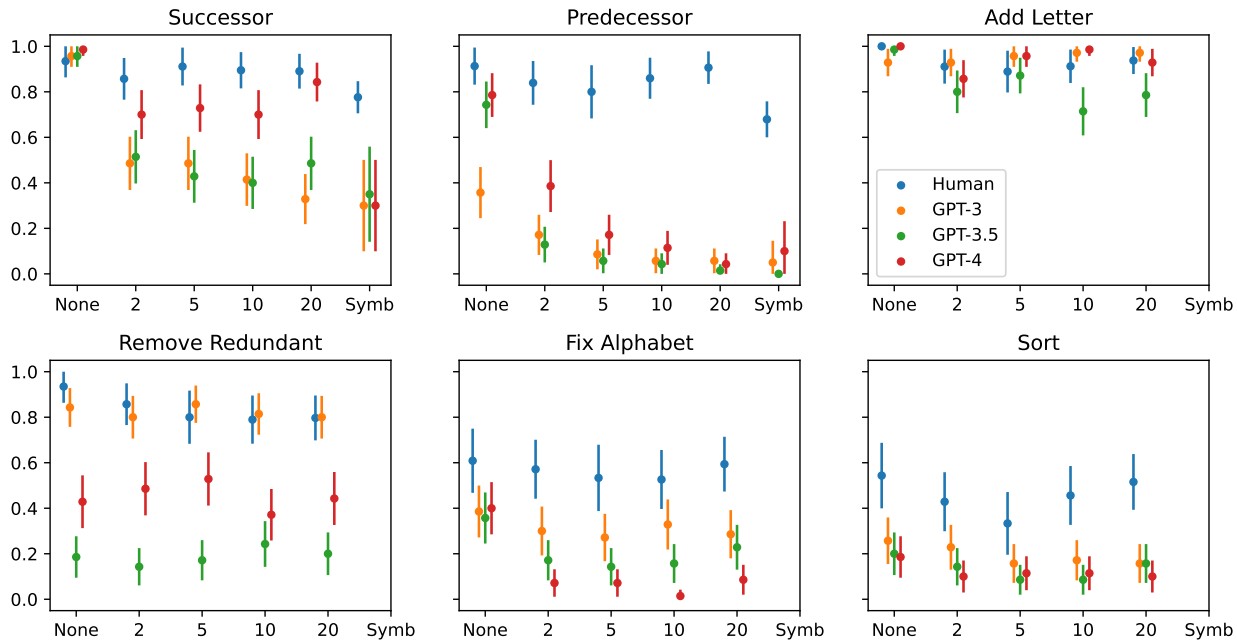

Figure 15: Performance of humans and GPT-models on zero-generalization tasks, broken down by problem type and number of letters permuted. Points are mean accuracy across samples and bars give 95% binomial confidence intervals. Numbers of samples for humans for each problem type are 46 for zero letters permuted, 56 for 2 letters permuted, 45 for 5 letters permuted, 57 for 10 letters permuted, 64 for 20 letters permuted, and 134 for symbolic alphabets. Samples per point for GPT models are 70 for each problem type and number of letters permuted from 0 to 20, and 40 for each Symb problem type (Successor and Predecessor). Figure is best viewed in color.

in this domain. However, there are other rules that can be inferred from a given pair of letter strings. We therefore examined the "incorrect" answers of humans and of GPT-3 and 4 to ascertain whether the kinds of errors made are similar.

For both GPT-3 and GPT-4, we randomly selected five incorrect answers from each problem type and alphabet, giving a sample of approximately 160 incorrect responses per GPT model. This number can be lower if there were fewer than 5 incorrect responses for a problem type and alphabet. For humans, we selected 184 incorrect answers.

| Problem Type | Source | Target | Literal Answer | Explanation |
|---|---|---|---|---|
| Succ | [f g h i] [f g h j] | [e f g h] | [e f g j] | Replace last letter with 'j'. |
| Fix | [b f g h i] [e f g h i] | [h i r k l] | [e i r k l] | Replace first letter with 'e'. |
| Rem | [g g h i j k] [g h i j k] | [k l m n n o] | [l m n n o] | Remove first letter of sequence. |
| Sort | [b c f e d] [b c d e f] | [v t u s w] | [v t w s u] | Swap 3rd and 5th letters. |

Table 6: Examples of literal interpretations of rules found by humans.

By manually examining these selections, we identified four broad categories of errors: 1) **Alternate rule formation**, where the answer given is consistent with an alternative rule. For example, if we have source transformation [a b c d] [a b c e] with target [i j k l], then according to the Successor rule the answer [i j k m] is correct. However, the answer [i j k e] is consistent with the rule "Replace the last letter with 'e'." 2) **Incorrect rule use**, in which the answer given is clearly related to the target letter string, and some kind of rule has been applied, but the rule is inconsistent with the example pair. For example, for [a b c d] [a b c e] with [i j k l], the response [i j k l m] is given. 3) **Wrong**, in which the answer given is inconsistent with the expected answer, but related to the target letter string. We could not discern any clear misunderstanding or alternate rule use. For example, for [a b c d] [a b c e] with [i j k l], the response [i j k q] is given. 4) **Completely Wrong**, in which the answer given is inconsistent with the expected answer, and unrelated to the target letter string. Again, we could not discern any clear misunderstanding or alternate rule use.

For example, for [a b c d] [a b c e] with [i j k l], the response [z y x b] is given. Table 7 gives percentages for each error type for humans and for each model. We see that in humans a large percentage (38.59%) of errors stem from using alternate rules. This is also seen in GPT-4 to a lesser extent (22%), but much less in GPT-3 (5.81%). We also see a difference in the percentage of incorrect rules applied, with GPT 3 and 4 both having over 30% of errors in this category and humans having around 15% of errors in this category. GPT models also have a higher percentage in the Wrong category, and for each of the models this category is the largest across the errors they made. Humans have a larger percentage of errors in the Completely Wrong category than do GPT-3 and 4 however. Across these four broad categories GPT-3 and 4 make different patterns of errors than humans.

Table 7: % error types across GPT-3, GPT-4, and Human

| | Error type | | | |
| --- | --- | --- | --- | --- |
| | Alt rule | Incorrect Rule | Wrong | Completely Wrong |
| GPT-3 | 5.81% | 30.97% | 55.48% | 7.74% |
| GPT-4 | 22.00% | 32.67% | 42.67% | 2.67% |
| Human | 38.59% | 14.67% | 34.24% | 12.50% |

We can further look at the kinds of alternative rules that are used by humans and by GPT. One key type of alternative rule is where a 'literal' interpretation of a rule is applied, illustrated in Table 6. As well as literal rules, humans found alternative rules for the Fix Alphabet problem type: they would interpret the changed letter as being moved a certain number of steps in the alphabet, and would move an equivalent letter in the prompt the same way. Usually "equivalent" means position; sometimes it means the identity of the letter. We find that GPT-4 gives the same kind of literal responses that humans do, but does not use alternative rules other than literal responses. GPT-3 has a limited number of errors in this category, and almost all are literal responses to Remove Redundant. In summary, within the "Alternative Rule" category, the GPT models found literal rules in the same way humans did, but did not find more inventive alternative rules.

Breaking down the Incorrect Rule category, we see more differences between human and GPT behavior. Human responses in this category are mostly where one of the rules has been applied in an incorrect situation, for example Add Letter has been applied instead of Successor. GPT-3 errors include adding two letters instead of one; continuing the alphabet; reversing the target; shifting the target; using an unpermuted alphabet instead of the one given; and repeating the target. GPT-4 made these mistakes and also generated responses that were too long. Very few humans made any of these mistakes. Out of the incorrect responses, the types of response made by humans and GPT models are very different.

## A.4 Additional Information on Digit-Matrix Experiments

### A.4.1 Digit Matrix Prompts

For our experiments on digit-matrix problems we experimented with several different prompts for GPT models. We found that the prompt described in Section 3.3 performed the best for all models; that was the prompt used for the results reported in Sections 3.4 and 3.5. The other prompts we tested were as follows:

**Alternate Prompt 1**

> **System:** You are a helpful assistant.
> **User:** [2] [2] [2]\n[5] [5] [5]\n[6] [6] [

**Alternate Prompt 2**

> **System:** You are a helpful assistant.
> **User:** Let's try to complete the pattern:\n\n[2] [2] [2]\n[5] [5] [5]\n[6] [6] [

**Alternate Prompt 3**

> **System:** You are a helpful assistant.
> **User:** Try to guess the missing piece. Give ONLY the answer with no explanation\n\n[2] [2] [2]\n[5] [5] [5]\n[6] [6] [

**Zero-Shot Chain-of-Thought Prompt**

> **System:** You are a genius at solving analogy problems.
> **User:** Try to complete the pattern below. Finish your answer with 'The answer is:' followed by the answer. \n\n[2] [2] [2]\n[5] [5] [5]\n[6] [6] [\nLet's think step by step.

**Zero-Shot Chain-of-Thought Prompt Alternate Blanks**

> **System:** You are a genius at solving analogy problems.
> **User:** Try to fill the gap in the pattern below. Finish your answer with 'The answer is:' followed by the answer. \n\n[2] [2] [ ]\n[5] [5] [5]\n[6] [6] [6]\nLet's think step by step.

## A.5 Digit Matrix Problems: More Detailed Results

Here we give more detailed results of human and GPT-model performance on our variants on digit-matrix problems. Figure 16b(a) gives the performance of our human participants on the original digit-matrix problems (blue points) and on the problems with alternate blank positions (orange points), for problems with different numbers of rules, and for logic problems. Human performance does not change significantly when alternate blank positions are used. Figure 16b(b) gives the performance of GPT-3.5 and GPT-4 on the original problems (blue and green points) and on problems with alternate blank positions (orange and red ploints). It can be seen that the performance of both models drops substantially for alternative blank positions in all cases.

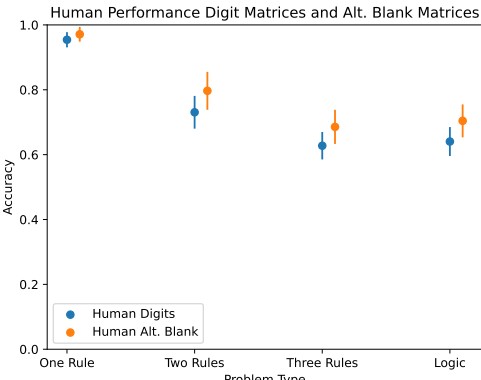
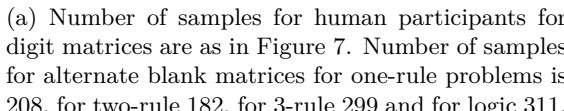
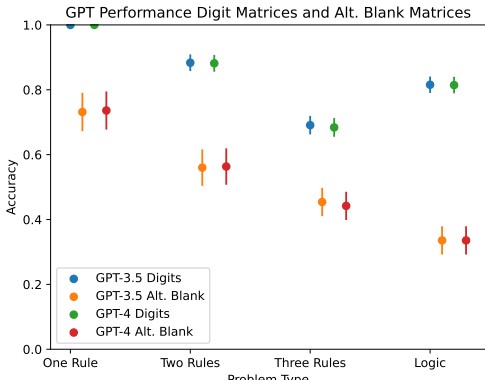

(a) Number of samples for human participants for digit matrices are as in Figure 7. Number of samples for alternate blank matrices for one-rule problems is 208, for two-rule 182, for 3-rule 299 and for logic 311.

(b) Numbers of samples for digit matrices are as in Figure 8. Number of samples for alternate blank matrices for one-rule problems is 216, for two-rule 300, for 3-rule 500 and for logic 450.

Figure 16: Accuracy on the original digit-matrix problems and those with and alternative blanks positions across different problem types for human participants and GPT models. Data points indicate mean accuracy across all problem types for each number of rules and bars indicate 95% binomial confidence intervals. Numbers of samples are given in individual captions. Figure is best viewed in color.

Figure 17(a) gives the performance of our human participants on the original digit-matrix problems (blue points) and on ones in which symbols replace numbers (orange points). The performance of humans does not show a substantial decrease when digits are replaced by symbols, except in the case of three-rule problems. Figure 17(b) shows a similar effect for GPT models, except in the case of logic problems, in which the performance of the GPT models does show a small but significant decrease when digits are replaced by symbols.

### A.5.1 Digit Matrices Zero-Shot Chain-of-Thought Prompt Results

We evaluated the performance of GPT-3.5 (0125) and GPT-4 (0613)[9] on the same digit matrices described in section 3.2. For the digit matrix and symbol matrix experiments we used the Zero-Shot Chain-of-Thought Prompt and for the

---

[9]GPT-3 was no longer available at the time of writing, and similarly gpt-3.5-turbo-0613 was no longer available for direct comparison with previous experiments.

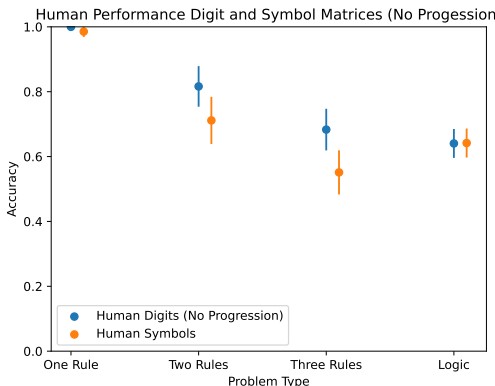
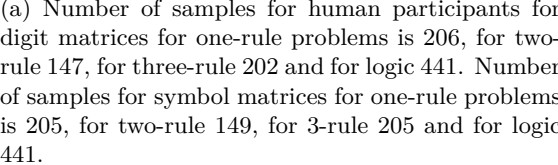
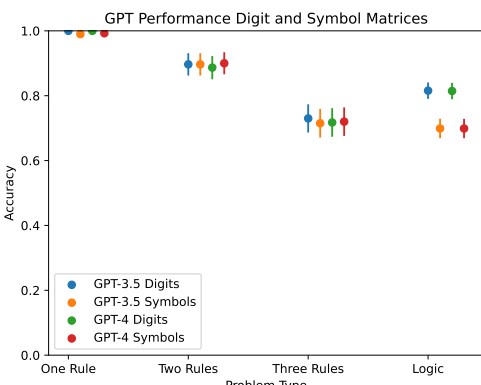

(a) Number of samples for human participants for digit matrices for one-rule problems is 206, for two-rule 147, for three-rule 202 and for logic 441. Number of samples for symbol matrices for one-rule problems is 205, for two-rule 149, for 3-rule 205 and for logic 441.

(b) Number of samples for both digit and symbol matrices for one-rule problems is 400, for two-rule 300, for 3-rule 400 and for logic 900.

Figure 17: Accuracy on the digit matrix and symbol matrix tasks across different numbers of rules, plus logic problems, for human participants and GPT models. Data points indicate mean accuracy across all task types for each number of rules and bars indicate 95% binomial confidence intervals. Numbers of samples are given in individual captions. Figure is best viewed in color.

alternate blank experiments, we used the Zero-Shot Chain of Thought Prompt Alternate Blanks given in Appendix A.4.1. All experiments were done with temperature set to zero and models were given 400 tokens to complete their response.

Results are reported in Figures 18 and 19. As the figures show, the zero-shot chain-of-thought prompt results in lower accuracy than our original prompt.

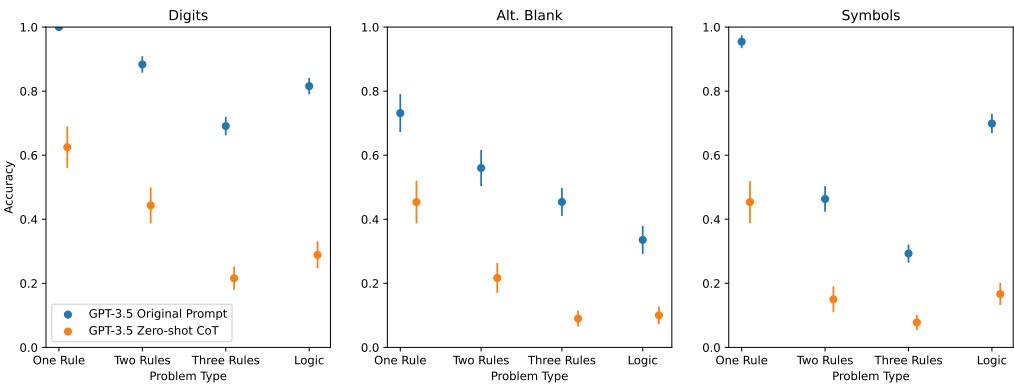

Figure 18: GPT 3.5, zero-shot chain-of-thought results: Accuracy and 95% binomial confidence intervals for the digit matrix task for GPT-3.5 (0125) with the original best performing prompt and zero-shot chain-of-thought prompting. Numbers of samples for either prompt are 2916 for Digits, 1,466 for Alt. Blank, and 2100 for Symbols. Figure is best viewed in color.

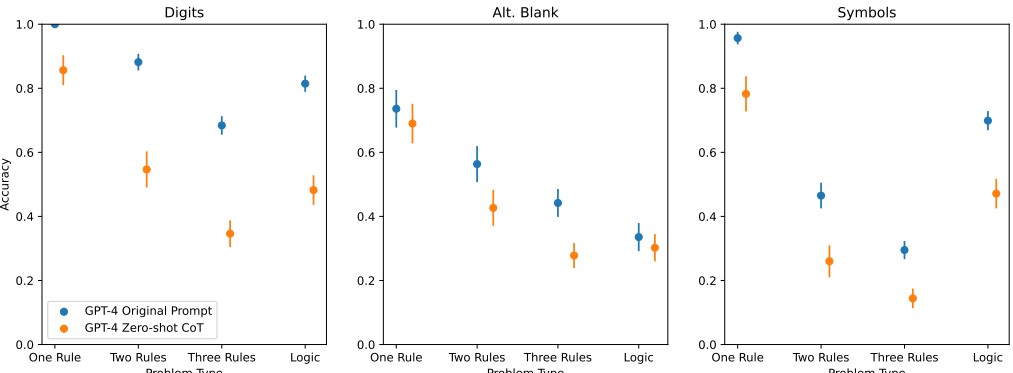

Figure 19: GPT-4, zero-shot chain-of-thought results; Accuracy and 95% binomial confidence intervals for the digit matrix task for GPT-4 (0613) with the original best performing prompt and zero-shot chain-of-thought prompting. Numbers of samples for either prompt are 2916 for Digits, 1,466 for Alt. Blank, and 2100 for Symbols. Figure is best viewed in color.

## A.6 Examples of Zero-Shot Chain-Of-Thought Reasoning for Digit Matrix Problems

The following are representative correct and incorrect responses from GPT-4 using zero-shot CoT prompting, as described in Section A.4.1.

**Digit Problem (correct response):**

```
[6 9] [8 9] [7 9]
[7 4] [6 4] [8 4]
[8 8] [7 8] [   ]
```

(correct answer: 6 8)

**Response:** In the first pair, the second number remains the same while the first number decreases by 1. In the second pair, the second number remains the same while the first number increases by 1.

So, following the same pattern, in the third pair, the second number should remain the same while the first number should decrease by 1.

The answer is: [6 8].

**Digit Problem (incorrect response):**

```
[6] [6] [6]
[3] [3] [3]
[1] [1] [ ]
```

(correct answer: 1)

**Response:** The pattern seems to be decreasing by half each time. The first row has 6, the second row has 3, which is half of 6. So, the third row should have half of 3, which is 1.5.

The answer is: [1.5]

**Digit Problem (incorrect response):**

```
[1 5] [3 8] [5 0]
[3 0] [5 5] [7 8]
[5 8] [7 0] [   ]
```

(correct answer: 9 5)

**Response:** First, let's look at the first numbers in each pair. They increase by 2 each time: 1, 3, 5, 3, 5, 7, 5, 7, and so on. So, the first number in the next pair should be 7.

Next, let's look at the second numbers in each pair. They follow a pattern of 5, 8, 0, 0, 5, 8, 8, 0, and so on. So, the second number in the next pair should be 5.

The answer is: [7 5]

**Alt. Blank Problem (correct response):**

```
[3 6] [2 7] [6 1]
[   ] [3 7] [2 1]
[2 6] [6 7] [3 1]
```

(correct answer: 6 6)

**Response:** First, let's look at the pattern in the first position of each bracket. In the first row, it goes from 3 to 2 to 6. In the third row, it goes from 2 to 6 to 3. It seems like the numbers are rotating in a cycle. If we follow this pattern, the first number in the second row should be 6.

Now, let's look at the pattern in the second position of each bracket. In the first row, it goes from 6 to 7 to 1. In the third row, it goes from 6 to 7 to 1 again. The pattern is the same in both rows. If we follow this pattern, the second number in the second row should be 6.

The answer is: [6 6]

**Alt Blank Problem (incorrect response):**

```
[0] [9] [ ]
[9] [4] [0]
[4] [0] [9]
```

(correct answer: 4)

**Response:** First, let's look at the pattern in the first two numbers of each row. In the first row, we go from 0 to 9, which is an increase of 9. In the second row, we go from 9 to 4, which is a decrease of 5. In the third row, we go from 4 to 0, which is a decrease of 4.

Now, let's look at the pattern in the last number of each row. In the first row, we don't know the last number yet. In the second row, the last number is 0. In the third row, the last number is 9.

So, the pattern seems to be that the last number of each row is the first number of the next row.

Therefore, the last number of the first row should be the first number of the second row, which is 9.

The answer is: 9.

**Symbol Problem (correct response):**

```
[& %] [< %] [∼ %]
[∼ (] [& (] [< (]
[< *] [∼ *] [   ]
```

(correct answer: & *)

**Response:** In the first set, the pattern moves from '&' to '<' to '∼'. In the second set, the pattern moves from '∼' to '&' to '<'. So, in the third set, the pattern should move from '<' to '∼' to '&'.

The answer is: [& *]

**Symbol Problem (incorrect response):**

```
[$] [!]  [∼]
```

```
[$] [!]  [∼]
[$] [!]  [ ]
```

(correct answer: ∼)

**Response:**  First, we can see that the pattern repeats every three symbols. The first symbol is always [$], the second symbol is always [!], and the third symbol alternates between [∼] and a blank space.

So, following this pattern, the third symbol in the third sequence should be a blank space.

The answer is: [$] [!] []

