# OpenReview forum: "Evaluating the Robustness of Analogical Reasoning in Large Language Models"
_TMLR — Accepted by TMLR_

### Review · Reviewer_nvRX · 2024-11-26

**Summary Of Contributions:**

This article explores the abstract reasoning abilities of LLMs through tasks adapted from Webb et al. (2023), including letter-string analogies, digit matrices, and story analogies. The findings reveal that, despite prior claims of LLMs excelling in zero-shot analogical reasoning, they fall short of the robustness demonstrated by humans in zero-shot analogy-making and show vulnerability to task variations.

**Audience:**

Yes

**Claims And Evidence:**

Yes

**Requested Changes:**

- On page 7, above Figure 3, the authors mention that "GPT-3.5 and GPT-4 show slightly lower accuracy than GPT-3, possibly due to their fine-tuning beyond a strict prompt-completion objective." Does this imply that the results are considered incorrect because the GPT answers are unparsable? Could the authors provide statistics on this issue, such as the ratio of unparsable results, and include case studies to illustrate the problem?

- It is observed that the results are sometimes presented as figures and other times as tables, even though they appear to convey the same information. Why not use a consistent format for all results?

- I suggest using different markers in the figures to represent different models, as the current color distinctions are hard to differentiate in black-and-white prints.

- It would be interesting to see how up-to-date open-source models such as LLaMA 3.2 or Gemma 2 perform on these tasks. I would anticipate that they perform similarly to the models already discussed, but presenting actual results would be beneficial. Additionally, since these models have both text-completion and instruction-tuned versions, the former might not encounter the issue mentioned in the first point.

- According to the template, the appendix should be put after the references.

**Strengths And Weaknesses:**

The paper is generally well-written, with concepts, background, and setups clearly introduced. It is easy to understand and follow. The results are reasonable, and the analysis of the experimental results is mostly well-grounded, with sufficient discussion of variations in human and GPT results. While the related work is not extremely comprehensive, it is adequate to support the claims made in the paper.

Regarding weaknesses, the main issue lies in the somewhat unsurprising conclusion. Many earlier works have reached the same conclusion using similar approaches (some of which are cited, others are not, though this does not significantly impact the paper). I have personally published and reviewed several similar works. Therefore, the conclusion may not be particularly exciting to the community, even though the evaluation method could be of interest to some. However, since the evaluation process is rigorous and reasonable, I don't think this should be a factor in rejecting the article, in accordance with the TMLR reviewing guidelines.

There are minor concerns and suggestions, which are detailed in the "Requested Changes" section below.

In general, I am inclined to accept this article.

---

> ### Author Response · Authors · 2025-02-02
> **Response to reviewer nvRX**
>
> Thank you for reviewing our paper. Here we respond to each of your comments and requests.
>
> **1. On page 7, above Figure 3, the authors mention that "GPT-3.5 and GPT-4 show slightly lower accuracy than GPT-3, possibly due to their fine-tuning beyond a strict prompt-completion objective." Does this imply that the results are considered incorrect because the GPT answers are unparsable? Could the authors provide statistics on this issue, such as the ratio of unparsable results, and include case studies to illustrate the problem?**  All results from GPT-3.5 and GPT-4  were parsable.  These results were given in the same format as those from GPT-3. We merely conjecture that the pattern completion abilities of later models may have been affected by differing training objectives.
>
> **2. It is observed that the results are sometimes presented as figures and other times as tables, even though they appear to convey the same information. Why not use a consistent format for all results?** In the revised paper we have converted some of the tables to be figures, to make the presentation consistent.  We kept tables where they seemed to us to be the most useful way of presenting results.
>
> **3. I suggest using different markers in the figures to represent different models, as the current color distinctions are hard to differentiate in black-and-white prints.** We considered using different markers but that made the plots look too cluttered.  So we kept the color distinctions, but added “Best viewed in color” to the figure captions, so that the reader is encouraged to look at the pdf with colors.  This seems to  be consistent with other published TMLR papers.
>
> **4. It would be interesting to see how up-to-date open-source models such as LLaMA 3.2 or Gemma 2 perform on these tasks. I would anticipate that they perform similarly to the models already discussed, but presenting actual results would be beneficial. Additionally, since these models have both text-completion and instruction-tuned versions, the former might not encounter the issue mentioned in the first point.** As we responded to reviewer SPSG, who made a similar suggestion, since our purpose is to test the robustness of earlier claims using GPT models, we did not include these more recent models.  Moreover, there is a reasonable chance that at least some of our problem variants are now in the training set of more recent models given that we have published preprints describing these variants, so to test fairly, we would have to generate wholly new sets of variants.
>
> In the revised paper, we have narrowed our description of our results to “GPT models” rather than “LLMs”, to reflect the scope of our claims.
>
> **5. According to the template, the appendix should be put after the references.** Done — see revised version.

---

### Review · Reviewer_SPSG · 2024-12-17

**Summary Of Contributions:**

The paper replicates and re-analyzes the robustness of Webb et al. (2023)'s claims of emergent analogy-making in GPT models. Their human study replication finds higher human performance in the same tasks as Webb et al. For the letter-string analogy task, GPT performance drops drastically when a different alphabet (permuted or symbols) is used, whereas human performance stays similar. For the digit matrix task, GPT performance drops significantly when the blank position is changed, but changing the symbols doesn't affect it, whereas human performance stays similar for both. For the story analogy task, a paraphrased version of the stories is designed which leads to drops in both human and GPT performance, but no statistically significant comparisons can be made.

**Audience:**

Yes

**Broader Impact Concerns:**

None.

**Claims And Evidence:**

No

**Requested Changes:**

1. Please add results on recent open-weight models like Llama 3.1 - 3.3, OLMO-2, Qwen 2.5 etc. and other API providers like Gemini Flash 2, Claude 3.5 Sonnet so that general claims about LLMs can be substantiated. This would also make this paper much more interesting to the audience if recent models are still struggling at these analogical reasoning tasks, confirming this testing didn't stop being interesting a year ago.  This would fix weaknesses A1, A2.

2. Please add examples to demonstrate the types of variants designed for the letter string analogy task. Also breakdown the aggregate results across the different variant types for a better understanding of whether certain types are leading to the performance drops. This would fix weakness B1.

3. Please also add results on few-shot examples, systematic prompting (using DSPy), and chain of thought to accurately report model performance. This would address weakness B2, B3.

4. Finally, it's unclear why human performance is higher in your replication than WHL's study. Could you add a discussion on the potential reasons for this?

**Strengths And Weaknesses:**

**Strengths**
1. Important replication study showing that GPT performance on cognitive tests may not be robust to perturbations which humans are insensitive to, such as changing the underlying alphabet in character level tasks.
2. Shows claims of superhuman analogical reasoning from a highly cited recent paper are questionable
3. Significant effort into replicating human studies, which proves fruitful as human performance is higher than the original study.

**Weaknesses**

A Only GPT-3, 3.5, 4 are studied. This leads to the following weaknesses:

1. Reproducibility may not be possible: The exact versions/dates of the models are not provided, so it is hard to reproduce these results as models keep changing, and also become unavailable after some time. This already affects their own study, "GPT-3 was not available for our experiments on symbol alphabets with one generalization".

2. Claims about "LLMs" cannot be made: In numerous occasions in the abstract / intro / conclusion, the paper generalises observations about GPT models studied to LLMs as a whole. This generalization cannot be made without studying other LLMs from other providers, including open-source ones.

B Concerns about experiment design:

1. Task descriptions for variants of letter-string analogies are unclear and written in abstract terms.
2. For the digit matrix task, putting the blank in other positions than the last one might make it harder for the model to identify what blank has to be filled. Humans are provided a clean 3x3 grid with a blank, whereas models are provided a linear string, which may not be a fair comparison and contribute to the drop in performance. Providing few shot examples can help fix this.
3. More generally, it is not clear why models are not allowed to use chain-of-thought, or few-shot-examples to understand the task format. Even humans are allowed some time to think and respond, and it's unfair to compare models given only 1 token of compute. Access to more inference-time compute can be essential to make the model expressive enough to solve certain kinds of reasoning tasks [1]. I understand this sticks to the WHL setup, but since the point of this paper is to argue that setup was flawed, it's worth also looking at ways in which model performance is underreported in that setup, not just overrated for an unbiased evaluation.

**References**

[1] Pfau, Jacob, William Merrill, and Samuel R. Bowman. "Let's Think Dot by Dot: Hidden Computation in Transformer Language Models." arXiv preprint arXiv:2404.15758 (2024).

---

> ### Author Response · Authors · 2025-02-02
> **Response to reviewer  SPSG.**
>
> Thank you for reviewing our paper. Here we respond to each of your comments and requests.
>
> **1. Claims about "LLMs" cannot be made: In numerous occasions in the abstract / intro / conclusion, the paper generalises observations about GPT models studied to LLMs as a whole. This generalization cannot be made without studying other LLMs from other providers, including open-source ones.**  In the revised version of the paper, in discussing our results we have replaced the general term “LLMs”  with the more specific “GPT models”, including in the title.
>
> **2. For the digit matrix task, putting the blank in other positions than the last one might make it harder for the model to identify what blank has to be filled. Humans are provided a clean 3x3 grid with a blank, whereas models are provided a linear string, which may not be a fair comparison and contribute to the drop in performance. Providing few shot examples can help fix this.**  As we discussed early in the paper, we wanted to stick with the original WHL zero-shot prompts.  But that being said, as is described in Section 3.5, we did test whether the GPT models “understood” which blank needed to be filled by giving digit matrices with the blank in each possible position and in each case asking  the models “What is the position of the missing element”?  GPT-4 was 100% accurate in giving the position of the missing element; GPT-3.5 was 65% accurate.  However, both models had similar low performance on solving the digit matrix tasks with alternate blank positions.  We concluded that because the models (or at least GPT-4) could reliably identify which blank has to be filled in, their low performance was not due to an inability to identify that blank.
>
> **3. More generally, it is not clear why models are not allowed to use chain-of-thought, or few-shot-examples to understand the task format. Even humans are allowed some time to think and respond, and it's unfair to compare models given only 1 token of compute. Access to more inference-time compute can be essential to make the model expressive enough to solve certain kinds of reasoning tasks [1]. I understand this sticks to the WHL setup, but since the point of this paper is to argue that setup was flawed, it's worth also looking at ways in which model performance is underreported in that setup, not just overrated for an unbiased evaluation.** In response to this comment and that of reviewer piBS, we performed a set of experiments using zero-shot CoT.  The letter-string experiments, results, and examples are in Appendix A1 of the revised paper (see Figure 14). The digit matrix experiments, results, and examples are in Appendix A2 of the revised paper (see Figures 18 and 19).  In short, we did not see any increased performance with zero-shot CoT; instead, adding “Let’s think step by step” to the prompt actually decreased the performance of both models we tested.
>
> We did not test giving few-shot examples, as this paper was meant to test the robustness of WHL’s results/claims, which are that emergent analogical reasoning abilities can be elicited with zero-shot queries.  (They emphasized that this is a key part of the claim: “Of particular interest is the ability of these models to reason about novel problems zero-shot, without any direct training. In human cognition, this capacity is closely tied to an ability to reason by analogy.”)
>
> **4. Please add results on recent open-weight models like Llama 3.1 - 3.3, OLMO-2, Qwen 2.5 etc. and other API providers like Gemini Flash 2, Claude 3.5 Sonnet so that general claims about LLMs can be substantiated. This would also make this paper much more interesting to the audience if recent models are still struggling at these analogical reasoning tasks, confirming this testing didn't stop being interesting a year ago. This would fix weaknesses A1, A2.** Since our purpose is to test the robustness of earlier claims using GPT models, we did not include these more recent models.  Moreover, there is a reasonable chance that at least some of our problem variants are now in the training set of more recent models given that we have published preprints describing these variants, so to test fairly, we would have to generate wholly new sets of variants.
>
> As we mentioned above, in the revised paper, we have narrowed our description of our results to “GPT models” rather than “LLMs”, to reflect the scope of our claims.
>
> **5. Please add examples to demonstrate the types of variants designed for the letter string analogy task. Also breakdown the aggregate results across the different variant types for a better understanding of whether certain types are leading to the performance drops. This would fix weakness B1.**  We give examples of variants in Figure 1 in the main paper.  Results across different variant types are given in Figure 5 and more in Figure 15 (in Appendix 2.3) of the revised paper.
>
> Responses continued in next comment.

---

> > ### Author Response · Authors · 2025-02-02
> > **Response to reviewer SPSG, continued.**
> >
> > **6. Please also add results on few-shot examples, systematic prompting (using DSPy), and chain of thought to accurately report model performance. This would address weakness B2, B3.** As we discuss in the response to #3 above, we have now included results of zero-shot CoT prompting.  For the reasons in that response, in this paper we decided to look only at zero-shot prompting.
> >
> > **7. Finally, it's unclear why human performance is higher in your replication than WHL's study. Could you add a discussion on the potential reasons for this?**  We do not know the reason for this. In section 2.4 we have this sentence speculating on potential reasons: “The differences on zero-generalization problems may be due to differences in experimental protocols or in the participant pools.”

---

### Review · Reviewer_piBS · 2024-12-26

**Summary Of Contributions:**

This paper evaluates three models of OpenAI's GPT family (GPT-3, GPT-3.5 Turbo and GPT-4) on a range of analogical reasoning tasks, with the goal of evaluating the robustness of findings from the Webb et al. paper ""Emergent analogical reasoning in large language models.". This previous paper had reported that GPT-3 generally matched our surpassed human participants in these abstract analogical reasoning tasks. This paper takes 3 such classes of tasks from Webb et al: letter-string analogies (completing sequences of letters following a pattern), digit matrices (completing a 2D matrix) and story analogies (associate similar causal relations between story elements). These tasks are augmented with several variants in order to probe the robustness of the models' reasoning capabilities. The findings can be summarized as: the paper largely replicates results with GPT-3 reported in Webb et al, but these results change substantially with the simple variants of the original tasks (e.g., using permuted alphabets), with LLM performance dropping more often (showing a lack of robustness) compared to human performance, which was generally higher. This was not the case in the story analogies, however, where human performance also dropped similarly after paraphrasing (but GPT-4 was more sensitive to answer order).

**Audience:**

Yes

**Claims And Evidence:**

No

**Requested Changes:**

* [minor, typo] intro: "and and variants"
* [minor, typo] paragraph title: "Results Oon the Two Types of Variant Problems"
* critical: please report experiments with CoT
* Please also add a few examples of raw model responses in the appendix, especially showing common error types. I appreciated the preliminary error analysis in A5, but think it would be enriched by having a few concrete examples besides just the statistics and descriptions of the errors.

**Strengths And Weaknesses:**

# Strengths

This paper studies an interesting question that has been the source of a lot of debate in the community: to what extent can LLMs reason, and how does their behavior compare to human reasoning. The focus on testing multiple variants of the same task for robustness is laudable. The authors also did several "sanity-checks" in each case, testing whether the LLMs understood the task even from the simple prompts that were given. I appreciate the reproduction of Webb et al. results with GPT-3 as a starting point, before moving on to evaluation on the new tasks. Generally, I think the paper is well-written and has a strong motivation, chooses interesting tasks and will provide a good starting point (tasks, human experiment results) for further work on these questions.

# Weaknesses

My main worry is about the methodology used to evaluate the LLMs. The authors main result is that the findings from Webb et al. break down once we look at simple variants of their task. From what I understand, the authors focus on what prior work calls "Direct Prediction", where the LLM outputs its answer without "thinking step-by-step" first. I believe the experiments in the paper do show clearly that this is not robust to intuitive variations of the original tasks. But I don't think that this is enough to say that the models cannot reason robustly, since this is just one way of trying to probe for this ability, and it's not the standard way anymore with modern models (chain-of-thought is). So, my main methodological criticism is that the authors should test their claims with zero-shot chain-of-thought instead: if they still hold, I believe the paper's argument will be much strengthened. I'll elaborate on the argument for why below.

In the introduction, Direct Prediction (DP) is motivated by the quote from WHL "Of particular interest is the ability of these models to reason about novel problems zero-shot, without any direct training". However, zero-shot chain-of-thought (prompting the model to reason first, without any task-specific guidance of examples) would still count as "zero-shot, no direct training", and it has been shown to elicit reasoning much more effectively than direct prediction. While it's essentially impossible to fully match the methodology between the human experiment and the LLM evaluation, DP seems to be more akin to giving participants a fixed, short time limit on each task, since the LLM can only do one forward pass to compute the answer token. CoT allows the model to think harder in harder instances, or instances that contradict the model's priors more (such as when changing the alphabet, or applying any other of the task generalizations that the authors test). Analogously, humans generally have higher response times in these instances - I don't know if the authors measured this, which could also be interesting to look at.

So to summarize this, I think that results on CoT would be important to make the claim that "these models often lack the zero-shot robustness of human analogy-making", which is the main stated claim of the paper. At the very least, you need to show in some tasks that CoT does not perform much different from DP in analogy making (which would be surprising but interesting, given the body of prior work on CoT observing otherwise) in order to mainly rely on the DP results to state this conclusion.

The paper also has some inconsistencies in which models are used in each experiment, due to text-davinci-003 having been phased out in early 2024, and some experiments were ran before that and some after (as clarified in Appendix A - though I was confused while reading the paper). It would be thus be impossible to run new CoT experiments on GPT-3 (this is not a fault of this paper, but is an issue in the general field of studying proprietary models in a way that makes replicating results later on impossible once the models are taken down).

---

> ### Author Response · Authors · 2025-01-22
> **Zero-shot CoT**
>
> Dear reviewer piBS,
>
> Thank you very much for your comments. We are currently in the process of running experiments with a zero-shot chain of thought prompt, appending 'Let's think step by step.' to the best performing prompt from our paper, as in https://arxiv.org/abs/2205.11916, and asking models to finish their response with 'The answer is: ' for ease of parsing. We are also giving models 400 tokens for their response which seems to be ample for them to finish reasoning (most responses use 50-100 tokens).
>
> We would like to confirm that this is the kind of setup you envisage.
>
> Many thanks!

---

> > ### Comment · Reviewer_piBS · 2025-01-22
> >
> > Dear authors,
> >
> > Thank you for the response and for setting up the new experiment. Yes, I think the setup you described would already be a fair comparison with zero-shot CoT, and if even CoT shows a similar lack of robustness this would significantly strengthen the argument in the paper. For the sake of understanding the results more qualitatively, it would also be great to see some representative examples of failures of CoT, perhaps in an Appendix or as the authors see fit in the discussion.
> >
> > Thanks!

---

> ### Author Response · Authors · 2025-02-02
> **Response to Reviewer piBS**
>
> Thank you for reviewing our paper.  Here we respond to each of your comments and requests.
>
> **1. The authors should test their claims with zero-shot chain-of-thought.**
> We performed these experiments.  The letter-string experiments, results, and examples are in Appendix A1 of the revised paper (see Figure 14). The digit matrix experiments, results, and examples are in Appendix A2 of the revised paper (see Figures 18 and 19).  In short, we did not see any increased performance with zero-shot CoT; instead, adding “Let’s think step by step” to the prompt actually decreased the performance of the models we tested.
>
> **2. [minor, typo] intro: "and and variants".**  Fixed.
>
> **3. [minor, typo] paragraph title: "Results Oon the Two Types of Variant Problems".**  Fixed.
>
> **4. Please also add a few examples of raw model responses in the appendix, especially showing common error types.** These have been added in appendices A1 and A2.

---

### Decision · Action_Editor_h69H · 2025-02-07

**Recommendation:** Accept as is

**Comment:**

The authors requested an extension to provide their rebuttal, which the AE approved. However, the system still requires the reviewers to make their recommendations earlier than the rebuttals were uploaded.
The AE carefully reads all reviewers' comments, the revision, and the authors' rebuttal. The AE believes that the main concerns (e.g. CoT prompting experiments, clarifications on the use of the zero-shot setup or GPT models) have been addressed by the rebuttal, and therefore is supportive of the submission to be accepted by TMLR.

**Audience:**

Yes. The paper is of interest to a reasonable number of people in TMLR's audience.

**Claims And Evidence:**

Yes. The claims made in the submission are supported by accurate, convincing and clear evidence.